# DENSITY CONSTRAINED REINFORCEMENT LEARNING

## ABSTRACT

Constrained reinforcement learning (CRL) plays an important role in solving safety-critical and resource-limited tasks. However, existing methods typically rely on tuning reward or cost parameters to encode the constraints, which can be tedious and tend to not generalize well. Instead of building sophisticated cost functions for constraints, we present a pioneering study of imposing constraints directly on the state density function of the system. Density functions have clear physical meanings and can express a variety of constraints in a straightforward fashion. We prove the duality between the density function and Q function in CRL and use it to develop an effective primal-dual algorithm to solve density constrained reinforcement learning problems. We provide theoretical guarantees of the optimality of our approach and use a comprehensive set of case studies including standard benchmarks to show that our method outperforms other leading CRL methods in terms of achieving higher reward while respecting the constraints.

## 1 INTRODUCTION

Constrained reinforcement learning (CRL) (Achiam et al., 2017; Altman, 1999; Dalal et al., 2018; Paternain et al., 2019; Tessler et al., 2019) has received increasing interests as a way of addressing the safety challenges in reinforcement learning (RL). CRL techniques aim to find the optimal policy that maximizes the cumulative reward signal while respecting the specified constraints. Existing CRL approaches typically involve constructing suitable cost functions and value functions to take into account the constraints. Then a crucial step is to choose appropriate parameters such as thresholds for the cost and value functions to encode the constraints. However, one significant gap between the use of such methods and solving practical RL problems is the correct construction of the cost and value functions, which is typically not solved systematically but relies on engineering intuitions (Paternain et al., 2019). Simple cost functions may not exhibit satisfactory performance, while sophisticated cost functions may not have clear physical meanings. When cost functions lack clear physical interpretations, it is difficult to formally guarantee the satisfaction of the performance specifications, even if the constraints on the cost functions are fulfilled. Moreover, different environments generally need different cost functions, which makes the tedious tuning process extremely time-consuming.

In this work, we fill the gap by imposing constraints on the *state density functions* as an intuitive and systematic way to encode constraints in RL. Density is a measurement of state concentration in the state space, and is directly related to the state distribution. It has been well-studied in physics (Yang, 1991) and control (Brockett, 2012; Chen & Ames, 2019; Rantzer, 2001). A variety of real-world constraints are naturally expressed as density constraints in the state space. Pure safety constraints can be trivially encoded as the entire density of the states being contained in the safe region. In more general examples, the vehicle densities in certain areas are supposed to be less than the critical density (Gerwinski & Krug, 1999) to avoid congestion. When spraying pesticide using drones, different parts of a farmland requires different levels of pesticide density. Indeed, in the experiments we will see how these problems are solved with guarantees using density constrained RL (DCRL).

Our approach is based on the new theoretical results of the duality relationship between the density function and value function in optimal control (Chen & Ames, 2019). One can prove generic duality between density functions and value functions for both continuous dynamics and discrete-state Markov decision processes (MDP), under various setups such as using Bolza form terminal constraints, infinite horizon discounted rewards, or finite horizon cumulative rewards. In Chen & Ames (2019) the duality is proved for value functions in optimal control, assuming that the full dynamics

of the world model is known. In this paper, we take a nontrivial step to establish the duality between the density function and the Q function (Theorem 1). We also reveal that under density constraints, the density function and Q function is also dual to each other (Theorem 2), which enables us to enforce constraints on state density functions in CRL. We propose a model-free primal-dual algorithm (Algorithm 1) to solve the DCRL problem, which is applicable in both discrete and continuous state and action spaces, and can be flexibly combined with off-the-shelf RL methods to update the policy. We prove the optimality of the policies returned by our algorithm if it converges (Proposition 1). We also discuss the approaches to computing the key quantities required by Algorithm 1.

**Our main contributions are:** 1) We are the first to introduce the DCRL problem with constraints on state density, which is associated with a clear physical interpretation. 2) We are the first to prove and use the duality between density functions and Q functions over continuous state space to solve DCRL. 3) Our model-free primal-dual algorithm solves DCRL and can guarantee the optimality of the reward and satisfaction of density constraints simultaneously. 4) We use an extensive set of experiments to show the effectiveness and generalization capabilities of our algorithm over leading approaches such as CPO, RCPO, and PCPO, even when dealing with conflicting requirements.

**Related work.** Safe reinforcement learning (García & Fernández, 2015) primarily focuses on two approaches: modifying the optimality criteria by combining a risk factor (Heger, 1994; Nilim & El Ghaoui, 2005; Howard & Matheson, 1972; Borkar, 2002; Basu et al., 2008; Sato et al., 2001; Dotan Di Castro & Mannor, 2012; Kadota et al., 2006; Lötjens et al., 2019) and incorporating extra knowledge to the exploration process (Moldovan & Abbeel, 2012; Abbeel et al., 2010; Tang et al., 2010; Geramifard et al., 2013; Clouse & Utgoff, 1992; Thomaz et al., 2006; Chow et al., 2018). Our method falls into the first category by imposing constraints and is closely related to constrained Markov decision processes (Altman, 1999) (CMDPs) and CRL (Achiam et al., 2017; Lillicrap et al., 2016). CMDPs and CRL have been extensively studied in robotics (Gu et al., 2017; Pham et al., 2018), game theory (Altman & Shwartz, 2000), and communication and networks (Hou & Zhao, 2017; Bovopoulos & Lazar, 1992). Most previous works consider the constraints on value functions, cost functions and reward functions (Altman, 1999; Paternain et al., 2019; Altman & Shwartz, 2000; Dalal et al., 2018; Achiam et al., 2017; Ding et al., 2020). Instead, we directly impose constraints on the state density function. Our approach builds on Chen et al. (2019) and Chen & Ames (2019), which assume known model dynamics. Instead, in this paper we consider the model-free setting and proved the duality of density functions to Q functions. In Geibel & Wysotzki (2005) density was studied as the probability of entering error states and thus has fundamentally different physical interpretations from us. In Dai et al. (2017) the duality was used to boost the actor-critic algorithm. The duality is also used in the policy evaluation community (Nachum et al., 2019; Nachum & Dai, 2020; Tang et al., 2019). The offline policy evaluation method proposed by Nachum et al. (2019) can also be used to estimate the state density in our paper, but their focus is policy evaluation rather than constrained RL. Therefore, we claim that this paper is the first work to consider density constraints and use the duality property to solve CRL.

## 2 PRELIMINARIES

**Markov Decision Processes (MDP).** An MDP $\mathcal{M}$ is a tuple $\langle S, A, P, R, \gamma \rangle$, where (1) $S$ is the (possibly infinite) set of states; (2) $A$ is the (possibly infinite) set of actions; (3) $P : S \times A \times S \mapsto [0, 1]$ is the transition probability with $P(s, a, s')$ the probability of transitioning from state $s$ to $s'$ when action $a \in A$ is taken; (4) $R : S \times A \times S \mapsto \mathbb{R}$ is the reward associated with the transition $P$ under the action $a \in A$; (5) $\gamma \in [0, 1]$ is a discount factor.

A policy $\pi$ maps states to a probability distribution over actions where $\pi(a|s)$ denotes the probability of choosing action $a$ at state $s$. Let a function $\phi : S \mapsto \mathbb{R}$ specifies the initial state distribution. The objective of an MDP optimization is to find the optimal policy that maximizes the overall discounted reward $J_p = \int_S \phi(s)V^\pi(s)ds$, where $V^\pi(s)$ is called the value function and satisfies $V^\pi(s) = r^\pi(s) + \gamma \int_A \pi(a|s) \int_S P(s, a, s')V^\pi(s')ds'da$, and $r^\pi(s) = \int_A \pi(a|s) \int_S P(s, a, s')R(s, a, s')ds'da$ is the one-step reward from state $s$ following policy $\pi$. For every state $s$ with occurring as an initial state with probability $\phi(s)$, it incurs a expected cumulative discounted reward of $V^\pi(s)$. Therefore the overall reward is $\int_S \phi(s)V^\pi(s)ds$. Although the equations are written in integral forms corresponding to continuous state-action space, the discrete counterparts can be derived similarly. Two major methods for solving MDPs are value iteration and

policy iteration, both based on the Bellman operator. Take value iteration as example, the Bellman optimality condition indicates $V(s) = \max_{a \in A} \int_S P(s, a, s')(R(s, a, s') + \gamma V(s'))ds'$. However, the formulation of value functions in MDP typically cannot handle constraints on the state distribution, which motivates the density functions.

**Density Functions.** Stationary state density functions $\rho_s : S \mapsto \mathbb{R}_{\geq 0}$ are measurements of the state concentration in the state space (Chen & Ames, 2019; Rantzer, 2001).[1] We will show later that generic duality relationship exists between density functions and value functions (or Q functions), which allows us to directly impose density constraints in RL problems. For infinite horizon MDPs, a given policy $\pi$ and an initial state distribution $\phi$, the stationary density of state $s$ is expressed as:

$$\rho_s^\pi(s) = \sum_{t=0}^{\infty} \gamma^t P(s^t = s | \pi, s^0 \sim \phi), \tag{1}$$

which is the discounted sum of the probability of visiting $s$. The key motivation behind the stationary density distribution is because most of the density constraints are time invariant and are instead defined over the state space. Stationary distribution gives a projection of all possible reachable states to the state space and is invariant over time. Therefore, as long as the stationary density satisfy the density constraints, the system always satisfies the density constraints at any time. It is straightforward to show that all theories and algorithms in this paper actually does not depend on the fact that the density distribution is stationary.

We prove in the appendix that the density has an equivalent expression

$$\rho_s^\pi(s) = \phi(s) + \gamma \int_S \int_A \pi(a|s') P(s', a, s) \rho_s^\pi(s') da ds', \tag{2}$$

where $\phi$ is the initial state distribution. $\phi$ coincides with the normalized (positive) supply function in Chen et al. (2019), which is defined as the rate of state $s$ entering the state space as an initial condition. We omit the details on constructing stationary distribution, which are provided in Chen et al. (2019).

**Definition 1** (DCRL). *Given an MDP $\mathcal{M} = \langle S, A, P, R, \gamma \rangle$ and an initial state distribution $\phi$, the density constrained reinforcement learning problem finds the optimal policy $\pi^\star$ that maximizes the expectation of the cumulative sum of a reward signal $\int_S \phi(s) V^\pi(s) ds$, subject to constraints on the stationary state density function represented as $\rho_{min}(s) \leq \rho_s^{\pi^\star}(s) \leq \rho_{max}(s)$ for $\forall s \in S$.*

## 3 DENSITY CONSTRAINED REINFORCEMENT LEARNING

In this section, we first show the duality between Q functions and density functions, and then introduce a primal-dual algorithm for solving DCRL problems.

### 3.1 DUALITY BETWEEN DENSITY FUNCTIONS AND Q FUNCTIONS

The duality relationship between density functions and value functions in optimal control is recently studied, for both dynamical systems and finite-state MDPs in Chen & Ames (2019). In this paper, we take a step further to show the duality property between the density function and Q function, for both continuous and discrete state MDPs. Our work is the *first* to prove and use the duality between density functions and Q functions over continuous state space to solve density constrained RL, which has not been explored and utilized by published RL literature. We use the standard infinite horizon discounted rewards case as an illustrative example. The duality in other setups such as finite horizon cumulative rewards and infinite horizon average rewards can be proved in a similar way.

To show the duality with respect to Q function, we extend the stationary density $\rho_s^\pi$ to consider the action taken at each state. Let $\bar{\rho}_s : S \times A \to \mathbb{R}_{\geq 0}$ be a stationary state-action density function, which represents the amount of state instances taking action $a$ at state $s$. $\bar{\rho}_s$ is related to $\rho_s$ via marginalization: $\rho_s(s) = \int_A \bar{\rho}_s(s, a) da$. Under a policy $\pi$, we also have $\bar{\rho}_s^\pi(s, a) = \rho_s^\pi(s)\pi(a|s)$.

---

[1]$\rho$ is not necessarily a probability density function. That is, $\int_S \rho(s) = 1$ is not enforced.

Let $r(s, a) = \int_S P(s, a, s') R(s, a, s') ds'$. Consider the density function optimization:

$$J_d = \max_{\bar{\rho}, \pi} \int_S \int_A \bar{\rho}_s^\pi(s, a) r(s, a) da ds$$

$$\text{s.t. } \bar{\rho}_s^\pi(s, a) = \pi(a|s) \left( \phi(s) + \gamma \int_S \int_A P(s', a', s) \bar{\rho}_s^\pi(s', a') da' ds' \right) \tag{3}$$

and the Q function optimization:

$$J_p = \max_{Q, \pi} \int_S \phi(s) \int_A Q^\pi(s, a) \pi(a|s) da ds$$

$$\text{s.t. } Q^\pi(s, a) = r(s, a) + \gamma \int_S P(s, a, s') \int_A \pi(a'|s') Q^\pi(s', a') da' ds' \tag{4}$$

**Theorem 1.** *The optimization objectives $J_d$ and $J_p$ are dual to each other and there is no duality gap. If both are feasible, then $J_d = J_p$ and they share the same optimal policy $\pi^\star$.*

The proof of Theorem 1 is included in the appendix. Theorem 1 states that optimizing the density function (dual) equals to optimizing Q functions (primal) since they will result in the same optimal policy $\pi^\star$. Since the dual optimization is directly related to density, we can naturally enforce density constraints on the dual problem. Consider the density constrained optimization:

$$J_d^c = \max_{\bar{\rho}, \pi} \int_S \int_A \bar{\rho}_s^\pi(s, a) r(s, a) da ds$$

$$\text{s.t. } \bar{\rho}_s^\pi(s, a) = \pi(a|s) \left( \phi(s) + \gamma \int_S \int_A P(s', a', s) \bar{\rho}_s^\pi(s', a') da' ds' \right) \tag{5}$$

$$\rho_{min}(s) \leq \rho_s^\pi(s) \leq \rho_{max}(s)$$

The marginalization $\rho_s^\pi(s) = \int_A \bar{\rho}_s^\pi(s, a) da$. Denote the Lagrange multipliers for $\rho_s^\pi \geq \rho_{min}$ and $\rho_s^\pi \leq \rho_{max}$ as $\sigma_- : S \mapsto \mathbb{R}_{\geq 0}$ and $\sigma_+ : S \mapsto \mathbb{R}_{\geq 0}$. The primal problem is formulated as:

$$J_p^c = \max_{Q, \pi} \int_S \phi(s) \int_A Q^\pi(s, a) \pi(a|s) da ds$$

$$\text{s.t. } Q^\pi(s, a) = r(s, a) + \sigma_-(s) - \sigma_+(s) + \gamma \int_S P(s, a, s') \int_A \pi(a'|s') Q^\pi(s', a') da' ds' \tag{6}$$

The difference between (6) and (4) is that the reward $r(s, a)$ is adjusted by $\sigma_-(s) - \sigma_+(s)$.

**Theorem 2.** *The density constrained optimization objectives $J_d^c$ and $J_p^c$ are dual to each other. If both are feasible and the KKT conditions are satisfied, then they share the same optimal policy $\pi^\star$.*

The proof of Theorem 2 is provided in the appendix. Theorem 2 reveals that when the KKT conditions are satisfied, the optimal solution to the adjusted unconstrained primal problem (6) is exactly the optimal solution to the dual problem (5) with density constraints. Such an optimal solution not only satisfies the state density constraints, but also maximizes the the total reward $J_d^c$. Thus it is the optimal solution to the DCRL problem.

**Remark 1.** The duality relationship between density and value function wide exists, for example, in Linear Programming. Technically, any nonnegative dual variable satisfying the conservation law (Liouville PDE in the continuous case, see Chen & Ames (2019)) is a valid dual. However, among all valid dual variables, the state density is associated with a clear physical interpretation as the concentration of states, and we are able to directly apply constraints on the density in RL.

**Remark 2.** Theorem 1 and 2 can be used in Constrained MDP (CMDP). In CMDP, the constraint $J_c = \int_S \mu(s) V_C(s) ds \leq \alpha$ is indeed a special case of the density constraint. $J_c = \int_S \mu(s) V_c(s) ds = \int_S \phi(s) V_c(s) ds = \int_S \rho_s(s) r_c(s) ds \leq \alpha \Leftrightarrow \int_S \rho_s(s) r_c(s) ds \leq \alpha$, where $r_c$ is the immediate cost and $V_c$ is the value of cost. Thus the CMDP constraint $J_c \leq \alpha$ equals to a special case of constraint on density $\rho_s$.

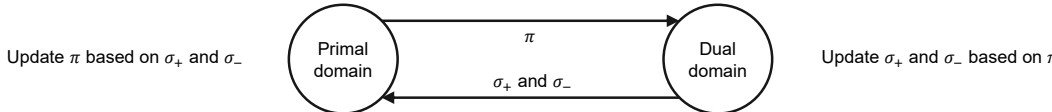

Figure 1: Illustration of the primal-dual optimization in DCRL. In the primal domain, we solve the adjusted primal problem in (6) to obtain the policy $\pi$. Then in the dual domain, the $\pi$ is used to evaluate the state density. Based on whether or not the density constraints and KKT conditions are satisfied, the Lagrange multipliers $\sigma_+$ and $\sigma_-$ are updated as $\sigma_+ \leftarrow \max(\mathbf{0}, \sigma_+ + \alpha(\rho_s^\pi - \rho_{max}))$ and $\sigma_- \leftarrow \max(\mathbf{0}, \sigma_- + \alpha(\rho_{min} - \rho_s^\pi))$. In the next loop, since the reward $r(s, a) + \sigma_-(s) - \sigma_+(s)$ is updated, the primal optimization solves for the new $\pi$ under the updated reward. The loop stops when the KKT conditions are satisfied.

## 3.2 PRIMAL-DUAL ALGORITHM FOR DCRL

The DCRL problem looks for an optimal policy of MDP subject to constraints on the density function. In this section, we continue considering the case where the stationary density in the infinite horizon discounted reward case must satisfy some upper and lower bounds: $\rho_{max}(s) \geq \rho_s(s) \geq \rho_{min}(s), \forall s \in S$. However, the primal-dual algorithm discussed in this section can be easily extended to handle other reward function setup and other types of constraints on the density function.

By utilizing the duality between density function and Q function (Theorem 2) in the density constrained optimization (5-6), the DCRL problem can be solved by alternating between the primal and dual problems, as is illustrated in Figure 1. In the primal domain, we solve the adjusted primal problem (reward adjusted by Lagrange multipliers) in (6) using off-the-shelf unconstrained RL methods such as TRPO (Schulman et al., 2015) and DDPG (Lillicrap et al., 2016). Note that the density constraints are enforced in dual domain and the primal domain is still an unconstrained problem, which means we can make use of existing RL methods to solve the primal problem. In the dual domain, the policy is used to evaluate the state density function, which is described in details in Section 3.3. If the KKT conditions $\sigma_+ \cdot (\rho_s^\pi - \rho_{max}) = 0$, $\sigma_- \cdot (\rho_{min} - \rho_s^\pi) = 0$ and $\rho_{min} \leq \rho_s^\pi \leq \rho_{max}$ are not satisfied, the Lagrange multipliers are updated and enter the next primal-dual loop. The key insight is that the density constraints can be enforced in the dual problem, and we can solve the dual problem by solving the equivalent primal problem using existing algorithms. Alternating between the primal and dual optimization can gradually adjust the Lagrange multipliers until the KKT conditions are satisfied. A general template of the primal-dual optimization with density constraint is provided in Algorithm 1.

---

**Algorithm 1** General template for the primal-dual optimization with density constraints

---

1: **Input** MDP $\mathcal{M}$, initial condition distribution $\phi$, constraints on the state density $\rho_{max}$ and $\rho_{min}$
2: Initialize $\pi$ randomly, $\sigma_+ \leftarrow \mathbf{0}$, $\sigma_- \leftarrow \mathbf{0}$
3: Generate experience $D_\pi \subset \{(s, a, r, s') \mid s^0 \sim \phi, a \sim \pi(s) \text{ then } r \text{ and } s' \text{ are observed}\}$
4: **Repeat**
5:     $(s, a, r, s') \leftarrow (s, a, r + \sigma_-(s) - \sigma_+(s), s')$ for each $(s, a, r, s')$ in $D_\pi$
6:     Update policy $\pi$ using $D_\pi$
7:     Generate experience $D_\pi \subset \{(s, a, r, s') \mid s^0 \sim \phi, a \sim \pi(s) \text{ then } r \text{ and } s' \text{ are observed}\}$
8:     Compute stationary density $\rho_s^\pi$ using $D_\pi$
9:     $\sigma_+ \leftarrow \max(\mathbf{0}, \sigma_+ + \alpha(\rho_s^\pi - \rho_{max}))$
10:     $\sigma_- \leftarrow \max(\mathbf{0}, \sigma_- + \alpha(\rho_{min} - \rho_s^\pi))$
11: **Until** $\sigma_+ \cdot (\rho_s^\pi - \rho_{max}) = 0$, $\sigma_- \cdot (\rho_{min} - \rho_s^\pi) = 0$ **and** $\rho_{min} \leq \rho_s^\pi \leq \rho_{max}$
12: **Return** $\pi$, $\rho_s^\pi$

---

In Lines 5-6 of Algorithm 1, the Lagrange multipliers $\sigma_+$ and $\sigma_-$ are used to adjust rewards, which lead to an update of the policy $\pi$. In Lines 7-8, the policy is used to evaluate the stationary density, then the Lagrange multipliers are updated following dual ascent (Lines 9-10). The iteration stops when all the KKT conditions are satisfied. Although Algorithm 1 is derived for the infinite horizon reward case, it also applies to the finite horizon case.

**Proposition 1.** *If Algorithm 1 converges to a feasible solution that satisfies the KKT condition, it is the optimal solution to the DCRL problem. (Proof provided in the appendix.)*

### 3.3 COMPUTATIONAL APPROACHES

Algorithm 1 requires computing the policy $\pi$, stationary density $\rho_s^\pi$, Lagrange multipliers $\sigma_+$ and $\sigma_-$. For $\pi$, there are well-developed representations such as neural networks and tabular methods. Updating $\pi$ using experience $D_\pi$ is also straightforward using standard approaches such as policy gradients or Q-Learning. By contrast, the computation of $\rho_s^\pi$, $\sigma_+$ and $\sigma_-$ is need to be addressed. The following computational approaches apply to both finite and infinite horizon.

**Density functions.** In the discrete state case, $\rho_s^\pi$ is represented by a vector where each element corresponds to a state. To compute $\rho_s^\pi$ from experience $D_\pi$ (line 8 of Algorithm 1), let $D_\pi$ contain $N$ episodes where episode $i$ ends at time $T_i$. Let $s^{ij}$ represent the state reached at time $j$ in the $i^{th}$ episode. Initialize $\rho_s^\pi \leftarrow \mathbf{0}$. For all $i \in \{1, \cdots, N\}$ and $j \in \{0, 1, \cdots, T_i\}$, do the update $\rho_s^\pi(s^{ij}) \leftarrow \rho_s^\pi(s^{ij}) + \frac{1}{N}\gamma^j$. The resulting vector $\rho_s^\pi$ approximates the stationary state density. In the continuous state space, $\rho_s^\pi$ cannot be represented as a vector since there are infinitely many states. We utilize the kernel density estimation method Chen (2017); Chen & Ames (2019) that computes $\rho_s^\pi(s)$ at state $s$ using the samples in $D_\pi$ with $\rho_s^\pi(s) = \frac{1}{N}\sum_{i=1}^N \sum_{j=0}^{T_i} \gamma^j K_h(s - s^{ij})$ where $K_h$ is the kernel function satisfying $\forall s \in S, K_h(s) \geq 0$ and $\int_S K_h(s)ds = 1$. There are multiple choices of the kernel $K_h$, e.g. Gaussian, Spheric, and Epanechnikov kernels Chen (2017), and probabilistic guarantee of accuracy can be derived, c.f. Wasserman (2019).

**Lagrange multipliers.** If the state space is discrete, both $\sigma_+$ and $\sigma_-$ are vectors whose length equals to the number of states. In each loop of Algorithm 1, after the stationary density is computed, $\sigma_+$ and $\sigma_-$ are updated following Line 9 and Line 10 respectively in Algorithm 1. If the state space is continuous, we construct Lagrange multiplier functions $\sigma_+$ and $\sigma_-$ from samples in the state space leveraging linear interpolation. Let $\bar{s} = [s_1, s_2, \cdots]$ represent the samples in the state space. In every loop of Algorithm 1, denote the Lagrange function computed by the previous loop as $\sigma_+^o$ and $\sigma_-^o$. We compute the updated Lagrange multipliers at states $\bar{s}$ as:

$$\bar{\sigma}_+ = [\max(0, \sigma_+^o(s_1) + \alpha(\rho_s^\pi(s_1) - \rho_{max}(s_1)), \max(0, \sigma_+^o(s_2) + \alpha(\rho_s^\pi(s_2) - \rho_{max}(s_2)), \cdots]$$
$$\bar{\sigma}_- = [\max(0, \sigma_-^o(s_1) + \alpha(\rho_{min}(s_1) - \rho_s^\pi(s_1)), \max(0, \sigma_-^o(s_2) + \alpha(\rho_{min}(s_2) - \rho_s^\pi(s_2)), \cdots]$$

Then the new $\sigma_+$ and $\sigma_-$ are obtained by linearly interpolating $\bar{\sigma}_+$ and $\bar{\sigma}_-$ respectively.

## 4 EXPERIMENT

We consider the MuJoCo (Todorov et al., 2012) benchmark and the autonomous electrical vehicle routing benchmark adopted from Blahoudek et al. (2020) in our experiment. We also report additional experimental results on 3 other benchmarks in the appendix, including a safe electrical motor control, an agricultural spraying drone, and a express delivery service transportation system, which all can show the power of our DCRL method over other approaches when dealing with complex density constraints. The evaluation criteria includes the *reward* and *constraint values*. The methods should keep the constraint values below the required thresholds and achieve as much reward as possible. The definition of reward and constraint vary from task to task and will be explained when each task is introduced. Our implementation will be made available upon acceptance of the paper.

**Baseline Approaches.** Three CRL baselines are compared. **PCPO** (Yang et al., 2020) first performs an unconstrained policy update then project the action to the constrained set. **CPO** (Achiam et al., 2017) maximizes the reward in a small neighbourhood that enforces the constraints. **RCPO** (Tessler et al., 2019) incorporates the cost terms and Lagrange multipliers with the reward function to encourage the satisfaction of the constraints. We used the original implementation of CPO and PCPO with KL-projection that leads to the best performance. For RCPO, since the official implementation is not available, we re-implemented RCPO and made sure it matches the original performance. RCPO restricts the expectation of the constraint values to be smaller than a threshold $\alpha$ instead of enforcing the constraints $\rho_{max}(s)$ and $\rho_{min}(s)$ for every state $s$. All the three baseline approaches and our DCRL have the same number of neural network parameters. Note that the baseline approaches enforce the constraints by restricting *return values* while our DCRL restricts the *state densities*. The constraint thresholds of DCRL and the baseline methods can be inter-converted by the duality of density functions and value functions (Chen & Ames, 2019).

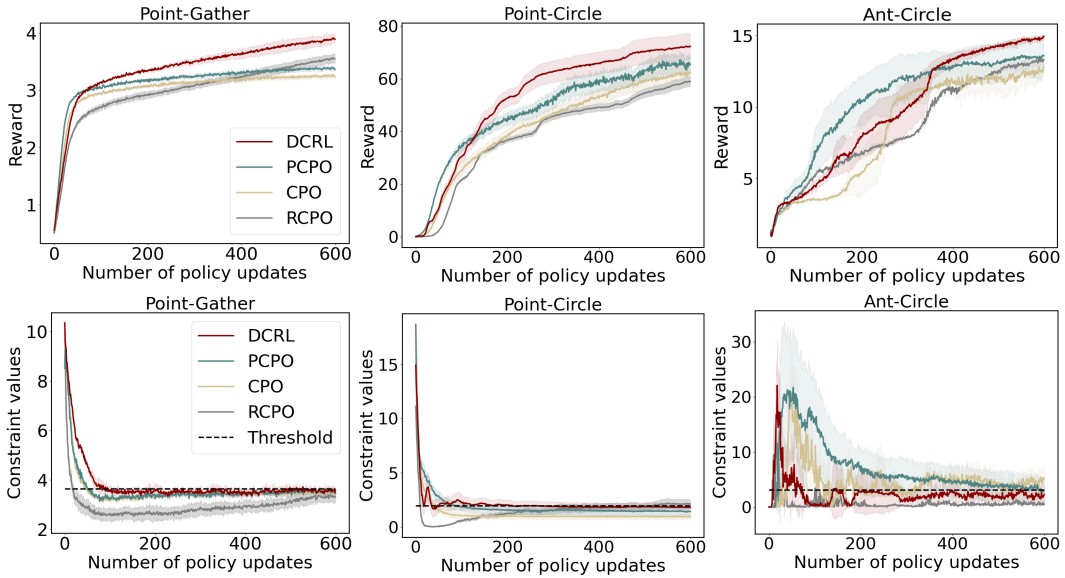

Figure 2: Performance on the constrained reinforcement learning tasks on the MuJoCo (Todorov et al., 2012) benchmark. All results are averaged over 10 independent trials. The methods are expected to achieve high reward while keeping the constraint values close to the threshold.

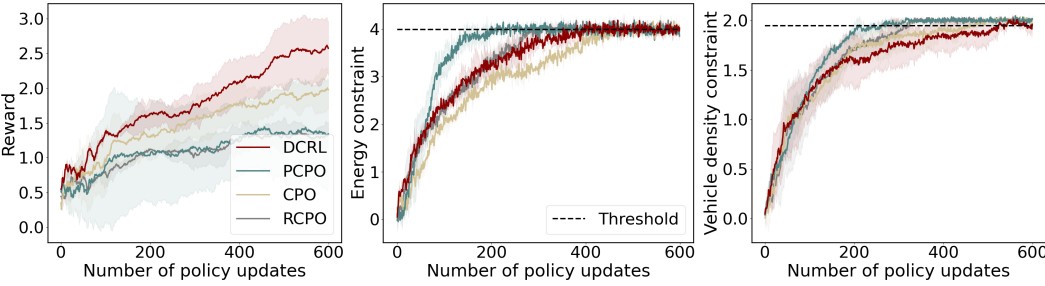

Figure 3: Performance on the autonomous electrical vehicle routing benchmark. All results are averaged over 10 independent trials. The methods are expected to keep close to the energy threshold (middle) and below the vehicle density threshold (right), while maximizing the reward (left).

## 4.1 MuJoCo Benchmark

**Domain Description.** Experiments are conducted on three tasks adopted from CPO (Achiam et al., 2017) on the MuJoCo benchmark, including Point-Gather, Point-Circle and Ant-Circle. In the Point-Gather task, a point agent moves on a plane and tries to gather as many apples as possible while keeping the probability of gather bombs below the threshold. In the Point-Circle task, a point agent tries to follow a circular path that maximizes the reward, while constraining the probability of exiting an given area. The Ant-Circle task is similar to the Point-Circle task except that the agent is an ant instead of a point. detailed configurations of the benchmarks can be found in Achiam et al. (2017).

**Experimental Results.** Figure 2 demonstrates the performance of the four methods. In general, DCRL is able to achieve higher reward than other methods while satisfying the constraint thresholds. In the Point-Gather and Point-Circle environments, all the four approaches exhibit stable performance with relatively small variances. In the Ant-Circle environment, the variances of reward and constraint values are significantly greater than that in Point environments, which is mainly due to the complexity of ant dynamics. In Ant-Circle, after 600 iterations of policy updates, the constraint values of the four approaches converge to the neighbourhood of the threshold. The reward of DCRL falls behind PCPO in the first 400 iterations of updates but outperforms PCPO thereafter.

### 4.2 AUTONOMOUS ELECTRICAL VEHICLE ROUTING BENCHMARK

**Domain Description.** Our second case study is about controlling autonomous electric vehicles (EV) in the middle of Manhattan, New York. It is adopted from Blahoudek et al. (2020) and is shown in Figure 4.2. When EVs drive to their destinations, they can avoid running out of power by recharging at the fast charging stations along the roads. At the same time, the vehicles should not stay at the charging stations for too long in order to save resources and avoid congestion. An road intersection is called a node. In each episode, an autonomous EV starts from a random node and drives to the goals. At each node, the EV chooses a direction and reaches the next node along that direction at the next step. The consumed electric energy is assumed to be proportional to traveling distance. The EV is fully charged at the beginning of each episode, and can choose to recharge at the fast charging stations. There are 1024 nodes and 137 charging stations in total. Denote the full energy capacity of the vehicle as $c_f$ and the remaining energy as $c_r$. When arriving at a charging station, the EV chooses a charging time $\tau \in [0, 1]$, then its energy increases to $\min(c_f, c_r + \tau c_f)$. The action space includes the EV direction and the charging time $\tau$. The state space $S \subset \mathbb{R}^3$ is consisted of the current 2D location and the remaining energy $c_r$.

Two types of constraints are considered: (1) the minimum remaining energy should keep close to a required threshold and (2) the vehicle density at charging stations should be less than a given threshold. Apparently, if the EV chooses a larger $\tau$, then the constraint (1) is more likely to be satisfied, while (2) is more likely to be violated, since a larger $\tau$ will increase the vehicle density at the charging station. These contradictory constraints pose a greater challenge to the RL algorithms. Both constraints can be naturally expressed as density constraints. For constraint (1), we can limit the density of low-energy states. For constraint (2), it is straightforward to limit the EV density (a function of $\mathbb{E}[\tau]$) at charging stations. The threshold of density constraints are transformed to the threshold of value functions to be used by the baseline methods. The conversion is based on the duality of density functions and value functions (Chen & Ames, 2019).

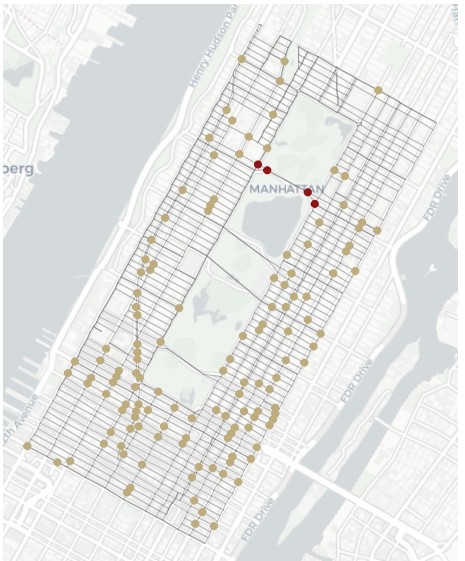

Figure 4: Autonomous electric vehicle routing in Manhattan: Control the electric vehicles to drive on the grey lines as roads and reach red nodes as goals. Vehicles can be charged at the gold nodes as fast charging stations. The roads and fast charging stations are from the real-world data (Blahoudek et al., 2020). More experimental results are provided in the appendix B.4.

**Experimental Results.** Figure 3 demonstrates the performance of the four approaches. Since this task has two contradicting constraints, it is challenging to satisfy both and at the same time maximize the reward. Although the baseline methods can approximately satisfy both constraints, their reward values are lower than the proposed DCRL method. Our Proposition 1 reveals that DCRL can achieve the optimal reward when enforcing the constraints, which is an important reason why DCRL shows a higher reward than other methods in this task.

## 5 CONCLUSION AND FUTURE WORKS

We introduced the DCRL problem to solve the RL problem while respecting the constraints on the state densities. State densities have clear physical meanings and can express a variety of constraints of the environment and the system. We proposed a model-free primal-dual algorithm to solve DCRL, which avoids the challenging problem of designing cost functions to encode constraints. Note that our algorithm does not guarantee the density constraints are satisfied in the learning process. In the future, we aim to improve the algorithm to enforce the density constraints during training as well. We also plan to identify the assumptions to prove the convergence of our algorithm.

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

## A    PROOFS OF STATEMENTS AND THEOREMS

In this section, the proofs omitted in our main paper are provided. We will prove the equivalent expression of the state density function, the duality between density function optimization and Q function optimization in the unconstrained case (Theorem 1) and constrained case (Theorem 2), as well as the optimality of the proposed primal-dual algorithm for DCRL (Proposition 1).

### A.1 Equivalent expression of the state density function

In Section 3.1, we point out that the state density function $\rho_s^\pi$ has two equivalent expressions. We start from Equation equation 1 and derive another expression:

$$
\begin{aligned}
\rho_s^\pi(s) &= \sum_{t=0}^\infty \gamma^t P(s^t = s|\pi, s^0 \sim \phi) \\
&= P(s^0 = s|\pi, s^0 \sim \phi) + \sum_{t=1}^\infty \gamma^t P(s^t = s|\pi, s^0 \sim \phi) \\
&= \phi(s) + \sum_{t=0}^\infty \gamma^{t+1} P(s^{t+1} = s|\pi, s^0 \sim \phi) \\
&= \phi(s) + \gamma \sum_{t=0}^\infty \gamma^t P(s^{t+1} = s|\pi, s^0 \sim \phi) \\
&= \phi(s) + \gamma \int_S \int_A \pi(a|s') P_a(s', s) \sum_{t=0}^\infty \gamma^t P(s^t = s'|\pi, s^0 \sim \phi) da ds' \\
&= \phi(s) + \gamma \int_S \int_A \pi(a|s') P_a(s', s) \rho_s^\pi(s') da ds'
\end{aligned}
$$

### A.2 Proof of Theorem 1

The Lagrangian of (3) is

$$
\mathcal{L} = \int_S \int_A r(s, a) \bar{\rho}_s^\pi(s, a) da ds -
$$
$$
\int_S \int_A \mu(s, a) \left( \bar{\rho}_s^\pi(s, a) - \pi(a|s)(\phi(s) + \gamma \int_S \int_A P_{a'}(s', s) \bar{\rho}_s^\pi(s', a') da' ds') \right) da ds \quad (7)
$$

where $\mu : S \times A \to \mathbb{R}$ is the Lagrange multiplier. The key step is by noting that

$$
\int_S \int_A \int_S \int_A \mu(s, a) \pi(a|s) P_{a'}(s', s) \bar{\rho}_s^\pi(s', a') da' ds' da ds \equiv
$$
$$
\int_S \int_A \int_S \int_A \mu(s', a') \pi(a'|s') P_a(s, s') \bar{\rho}_s^\pi(s, a) da' ds' da ds \quad (8)
$$

Then by rearranging terms, the Lagrangian becomes

$$
\mathcal{L} = \int_S \phi(s) \int_A \mu(s, a) \pi(a|s) da ds -
$$
$$
\int_S \int_A \bar{\rho}_s^\pi(s, a) \left( \mu(s, a) - r(s, a) - \gamma \int_S P_a(s, s') \int_A \pi(a'|s') \mu(s', a') da' ds' \right) da ds \quad (9)
$$

By the KKT condition and taking $Q = \mu$, the optimality condition satisfies equation 4 exactly, and when optimality condition is attained, $J_d^\star = \mathcal{L} = J_p^\star$.

### A.3 Proof of Theorem 2

The solution $\pi$ to the primal problem is the optimal policy for the modified MDP with reward $r + \sigma_- - \sigma_+$, which means $\pi$ is the optimal solution to:

$$
\max_{Q,\pi} \int_S \phi(s) \int_A Q^\pi(s, a) \pi(a|s) da ds \quad (10)
$$
$$
\text{s.t. } Q^\pi(s, a) = r(s, a) + \sigma_-(s) - \sigma_+(s) + \gamma \int_S P_a(s, s') \int_A \pi(a'|s') Q^\pi(s', a') da' ds' \quad (11)
$$

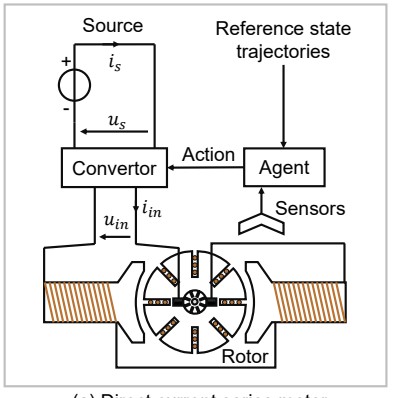
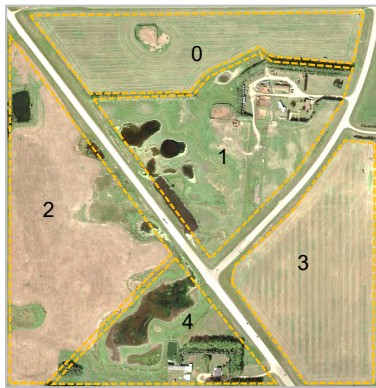

(a) Direct current series motor       (b) A farmland in Saskatchewan, Canada

Figure 5: Illustration of the electrical motor control and drone application environments. (a) Electrical motor control: Control the motor to follow the reference trajectories and avoid motor overheating. (b) Agricultural pesticide spraying: Control the drones to spray pesticide over a farmland which is divided into five parts and each requires different densities of pesticide.

According to Theorem 1, $\pi$ is also the optimal solution to:

$$\max_{\bar{\rho},\pi} \int_S \int_A \bar{\rho}_s^\pi(s,a)(r(s,a) + \sigma_-(s) - \sigma_+(s))dads \tag{12}$$

$$\text{s.t. } \bar{\rho}_s^\pi(s,a) = \pi(a|s)\left(\phi(s) + \gamma \int_S \int_A P_{a'}(s',s)\bar{\rho}_s^\pi(s',a')da'ds'\right) \tag{13}$$

Therefore, for any feasible policy $\pi'$ the following inequality holds:

$$\int_S \int_A \bar{\rho}_s^\pi(s,a)(r(s,a) + \sigma_-(s) - \sigma_+(s))dads \geq$$
$$\int_S \int_A \bar{\rho}_s^{\pi'}(s,a)(r(s,a) + \sigma_-(s) - \sigma_+(s))dads \tag{14}$$

By complementary slackness, if $\sigma_-(s) > 0$, then $\rho_s^\pi(s) = \rho_{min}(s)$. The same applies to $\sigma_+$ and $\rho_{max}$. Since $\rho_s^{\pi'}(s) \geq \rho_{min}(s)$ and $\rho_s^{\pi'}(s) \leq \rho_{max}(s)$, we have:

$$\int_S \int_A \bar{\rho}_s^\pi(s,a)(\sigma_-(s) - \sigma_+(s))dads \leq \int_S \int_A \bar{\rho}_s^{\pi'}(s,a)(\sigma_-(s) - \sigma_+(s))dads \tag{15}$$

Then we use equation 15 to eliminate the $\sigma_-(s) - \sigma_+(s)$ in (14) and derive:

$$\int_S \int_A \bar{\rho}_s^\pi(s,a)r(s,a)dads \geq \int_S \int_A \bar{\rho}_s^{\pi'}(s,a)r(s,a)dads \tag{16}$$

which means $\pi$ is the optimal solution maximizing $J_d^\star$ among all the solutions satisfying the density constraints. As a result, $\pi$ is the optimal solution to the DCRL problem.

### A.4 PROOF OF PROPOSITION 1

Note that when Algorithm 1 converges and the KKT conditions are satisfied, the policy that Algorithm 1 founds is the optimal solution to the primal problem (6) because the algorithm explicitly solves (6) in Line 6. Thus, by Theorem 2, the policy is the optimal solution to the DCRL problem.

## B SUPPLEMENTARY EXPERIMENTS

In this section, we provide new case studies that are not covered in the main paper. We mainly compare with RCPO (Tessler et al., 2019) and the unconstrained DDPG (Lillicrap et al., 2016), which serves as the upper bound of the reward that can be achieved if the constraints are ignored.

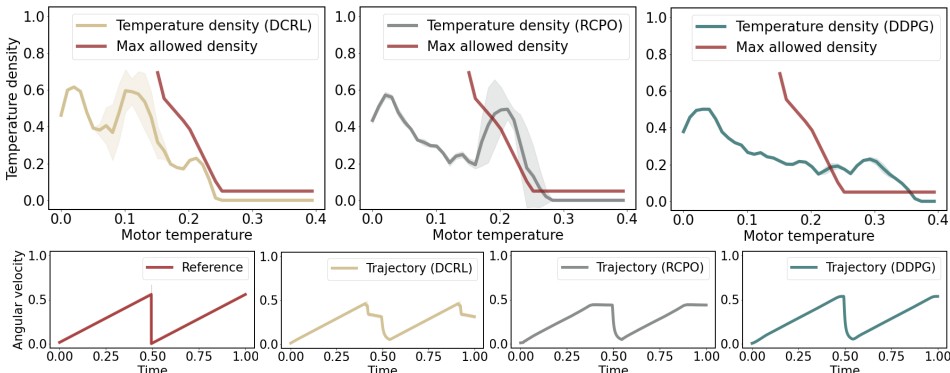

Figure 6: Density curves of the motor's temperature when following sawtooth-wave trajectories using different methods. The temperature is relative to and also normalized using the environment temperature.

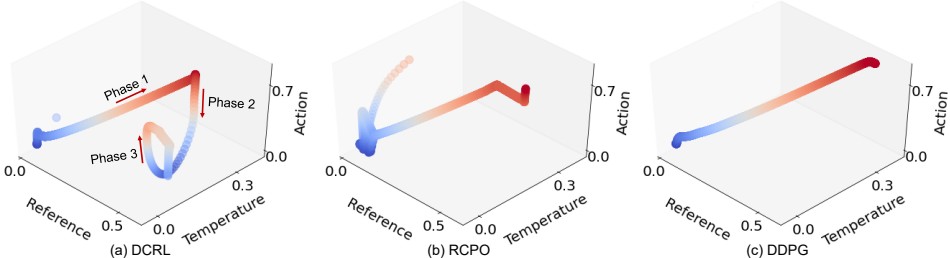

Figure 7: Visualization of the behavior of three methods in the safe electrical motor control task.

### B.1 SAFE ELECTRICAL MOTOR CONTROL

The safe electrical motor control environment is from Traue et al. (2019) and is illstrated in Figure 5 (a). The objective is to control the motor so that its states follow a reference trajectory and at the same time to prevent the motor from overheating. The state space $S \subset \mathbb{R}^6$ and consists of six variables: angular velocity, torque, current, voltage, temperature and the reference angular velocity. The action space $A \subset \mathbb{R}$ is the electrical duty cycle that controls the motor power. The agent outputs a duty cycle at each time step to drive the angular velocity close to the reference. When the reference angular velocity increases, the required duty cycle will need to increase. As a result, the motor's power and angular velocity will increase and cause the motor temperature to grow.

The algorithms are trained and tested using sawtooth-wave reference trajectories. Results for other types of trajectories are presented in the supplementary material. Figure 6 shows that our approach successfully controls the temperature density below the given threshold. Unconstrained DDPG does not consider the temperature and only minimizes the difference between reference and measured angular velocity. Therefore, the motor controlled using unconstrained DDPG has a significant violation of the required temperature density when the motor temperature is high. RCPO manages to avoid the high temperature but fails to suppress the density completely below the threshold. As a comparison, our method manages to successfully control the temperature density completely below the required threshold.

To gain some insight on the different performance of the three methods, we visualize the trajectories and actions (duty cycles) taken at different temperatures and reference angular velocities in Figure 7. In Figure 7 (a), The trajectory using DCRL can be divided into three phases. In Phase 1, as the reference angular velocity grows, the duty cycle also increases, so the motor temperature goes up. When the temperature is too high, the algorithm enters Phase 2 where it reduces the duty cycle to control the temperature, even though the reference angular velocity remains high. As the temperature goes down, the algorithm enters Phase 3 and increases the duty cycle again to drive the motor angular velocity closer to the reference. In Figure 7 (b), when the temperature is high, the RCPO algorithm will stop increasing the duty cycle but will not decrease it as Algorithm 1 does. So the temperature remains high and thus the density constraints are violated. In Figure 7 (c), the unconstrained DDPG algorithm continues to increase the duty cycle in spite of the high temperature.

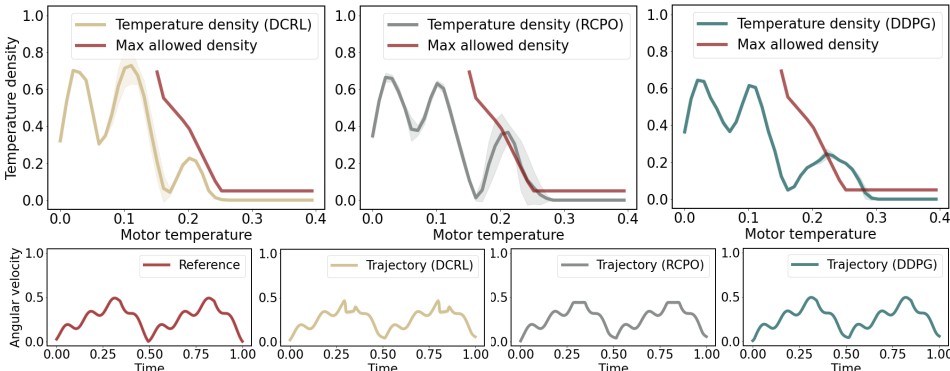

Figure 8: Density curves of the motor's temperature when following asymmetric sine-wave trajectories using different methods, which are trained with sawtooth-wave trajectories. None of the methods have seen the asymmetric sine-wave trajectories during training. The temperature is relative to and also normalized using the environment temperature. DDPG almost perfectly follows the angular velocity trajectory but violates the density constraints on high-temperature states. RCPO slightly violates the density constraints. DCRL is able to follow most part of the trajectories and completely satisfy the constraints.

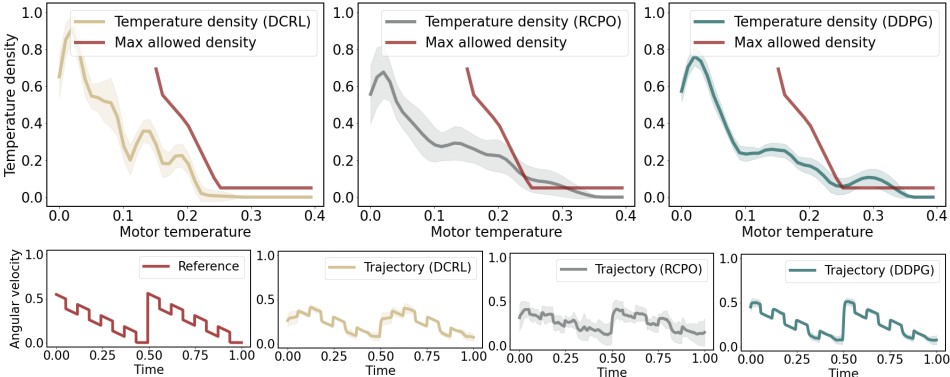

Figure 9: Density curves of the motor's temperature using different methods trained and tested with staircase-wave trajectories. The temperature is relative to and also normalized using the environment temperature. The unconstrained DDPG violates the temperature constraints while perfectly following the trajectory. RCPO is better than the unconstrained DDPG in terms of restricting the temperature, but its angular velocity trajectory is not as stable as DDPG. DCRL can successfully control the temperature to meet the constraints.

To examine the generalization capability of our method, here we train on the sawtooth-wave trajectories but test on asymmetry sine trajectories that were not seen by Algorithm 1 before (Figure 8). To make the experiments more diverse, we also train and test the methods using staircase-wave trajectories (Figure 9). Note that all the configurations of our DCRL method are the same in the three experiments (Figure 6, 8-9) and we did not tune any of the hyper-parameters for each type of reference trajectory.

In order to demonstrate the difficulty of tuning cost functions to satisfy the constraints with the RCPO method, we present the results of RCPO with different hyper-parameter configurations of the cost function (Figure 10). Throughout the experiment, RCPO uses the cost function $\max\{0, (s \downarrow t) - \eta\}$, where $s \downarrow t$ is the temperature variable under state $s$. The cost function is positive when $s \downarrow t$ exceeds $\eta$. $\eta = 0.20$ for RCPO, $\eta = 0.23$ for RCPO-V2, $\eta = 0.17$ for RCPO-V3 and $\eta = 0.16$ for RCPO-V4. For fair comparisons, other parameters of these three variants of RCPO remain unchanged. It is clear that RCPO-V4 is only slightly different from RCPO-V3.

As is shown in Figure 8, even if the testing reference trajectories are unseen, our approach still manages to control the temperature density below the threshold. The sawtooth-wave trajectories are piece-wise linear, which are for training. By contrast, the asymmetric sine-wave trajectories are nonlinear curves, which are for testing. Figure 8 shows the generalization capability from linear to nonlinear reference trajectories without any re-training. In Figure 9, while the angular velocity trajectory of RCPO shows unstable high-frequency oscillation, DCRL can stabilize the angular

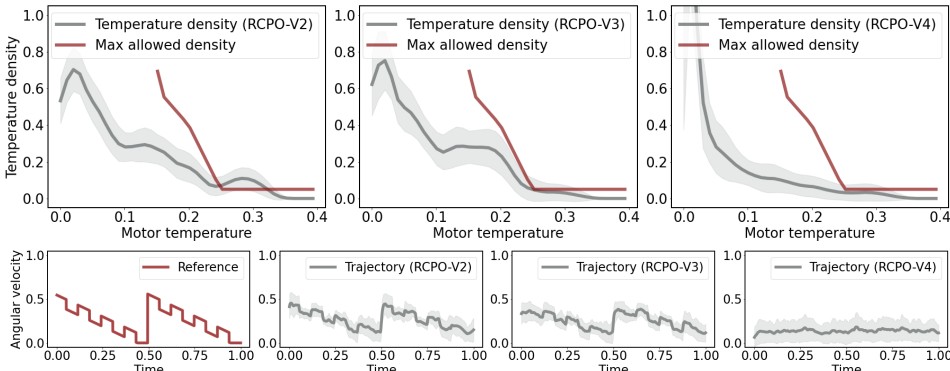

Figure 10: Density curves of the motor's temperature when following staircase-wave trajectories using the RCPO method with different configurations of the cost function. It is still possible to achieve satisfactory performance (RCPO-V3) through extensive cost function tuning. However, though RCPO-V4 is only slightly different from RCPO-V3 in terms of hyper-parameters, it completely fails to follow the reference trajectory. This observation suggests that it could be difficult to design and tune the cost functions to satisfy the density constraints.

velocity and keep the temperature density completely below the threshold. Unconstrained DDPG does not take into account the temperature density constraints and thus causes the motor to overheat. All these scenarios show that our method can constantly outperforms RCPO and the unconstrained DDPG in terms of guaranteeing satisfaction of constraints and generalization capability.

Different configurations in the cost (negative reward) function of RCPO will yield different results, and we show that it could be difficult to tune the cost functions to satisfy the density constraints. In Figure 10, though it is still possible for RCPO to satisfy the density constraints by tuning the hyper-parameters of the cost function (as RCPO-V3), a small perturbation would have an unexpected negative effect. The hyper-parameter of RCPO-V4 is only slightly different from that of RCPO-V3, but RCPO-V4 completely fails to follow the reference trajectory. A larger $\eta$ will incur a heavier punishment for high temperature states, but tuning $\eta$ could be difficult. Moreover, when the reference trajectory changes, the feasible $\eta$ also changes, making the tuning process time-consuming. In comparison, we did not tune any parameter of the DCRL method for each reference trajectory, which indicates the robustness and generalization capability of DCRL when applied to new scenarios.

## B.2 AGRICULTURAL SPRAYING DRONE

We consider the problem of using drones to spray pesticides over a farmland in simulation. Consider the area in Figure 5 (b). The drone starts from the top-left corner, flies over the farmland spraying pesticides, and stops at the bottom-right corner. At each timestep, the agent outputs an acceleration parallel to the ground and then the drone moves horizontally at a fixed height. The drone has a constant spraying speed, which means when the drone moves faster, there would be less pesticides on the land that the drone passes over. At each moment, the pesticides sprayed by the drone uniformly cover an area centered at the drone with diameter equals to 0.02 times the side length of the farmland. Two constraints are (1) the pesticide density of each part of the land is within the predefined minimum and maximum pesticide density requirements, and (2) the density (distribution) of drone velocity is below a predefined threshold, which is to prevent dangerous high-speed movement.

As is shown in Figure 11, for each method we evaluate the area that satisfies pesticide density constraints, the time consumption and the velocity density. While RCPO and our method demonstrate similar performance in terms of controlling the pesticide density and velocity density, our method requires fewer time steps to finish the task. The unconstrained DDPG algorithm takes less time, but cannot satisfy the requirement on pesticide density or velocity density. More experiments on the agricultural spraying problem are presented in the supplementary material.

We also examine the methods with different pesticide density requirements to assess their capability of generalizing to new scenarios. For the farmland shown in Figure 5 (b), from area 0 to 4, the minimum and maximum pesticide density are $(1, 0, 0, 1, 1)$ and $(2, 0, 0, 2, 2)$ respectively. In this supplementary material, we evaluate with two new configurations. In Figure 12, the minimum and maximum density are set to $(0, 1, 1, 0, 1)$ and $(0, 2, 2, 0, 2)$ from area 0 to 4. In Figure 13, the

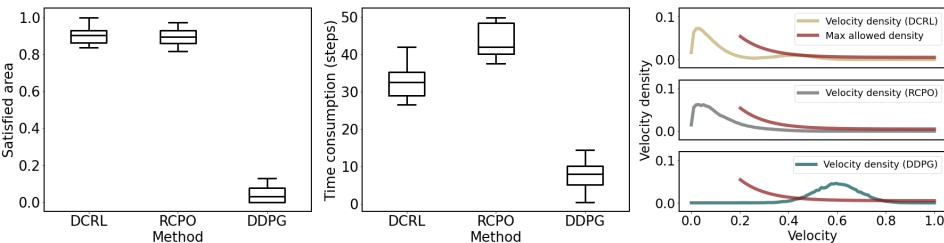

Figure 11: Results of the agricultural spraying problem. Left: Percentage of the entire area that satisfies the pesticide density requirement. Middle: Time consumption in steps. Whiskers in the left and middle plots denote confidence intervals. Right: visualization of the velocity densities using different methods.

minimum and maximum density are set to $(0, 0, 1, 1, 0)$ and $(0, 0, 2, 2, 0)$ from from area $0$ to $4$. Other configurations remain the same.

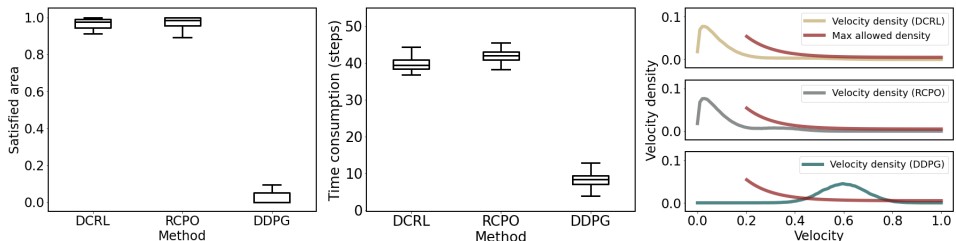

Figure 12: Results of the agricultural spraying problem with minimum pesticide density $(0, 1, 1, 0, 1)$ and maximum density $(0, 2, 2, 0, 2)$ from area $0$ to $4$. Left: Percentage of the entire area that satisfies the pesticide density requirement. Middle: Time consumption in steps. Whiskers in the left and middle plots denote confidence intervals. Right: visualization of the velocity densities using different methods.

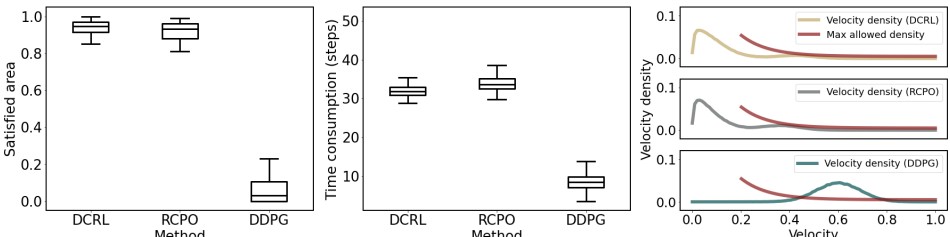

Figure 13: Results of the agricultural spraying problem with minimum pesticide density $(0, 0, 1, 1, 0)$ and maximum density $(0, 0, 2, 2, 0)$ from area $0$ to $4$. Left: Percentage of the entire area that satisfies the pesticide density requirement. Middle: Time consumption in steps. Whiskers in the left and middle plots denote confidence intervals. Right: visualization of the velocity densities using different methods.

In Figure 12 and 13, DCRL and RCPO demonstrates similar performance in controlling pesticide densities to be within the minimum and maximum thresholds, while DCRL demands less time to finish the task. DDPG only minimizes the time consumption and thus requires the least time among the three methods, but cannot guarantee the pesticide density is satisfied. In terms of the velocity control, both DCRL and RCPO can avoid the high-speed movement. These observations suggest that when both DCRL and RCPO finds feasible policies satisfying density constraints, the policy found by DCRL can achieve lower cost or higher reward defined by the original unconstrained problem, which is the time consumption of executing the task in this case study.

## B.3 EXPRESS DELIVERY SERVICE TRANSPORTATION

An express delivery service company has several service points and a ship center in a city. An example configuration is illustrated in Figure 14 (a). The company uses vans to transport the packages from each service point to the ship center. The vans start from some service points following an initial distribution, travel through some service points and finally reach the ship center. The cost is

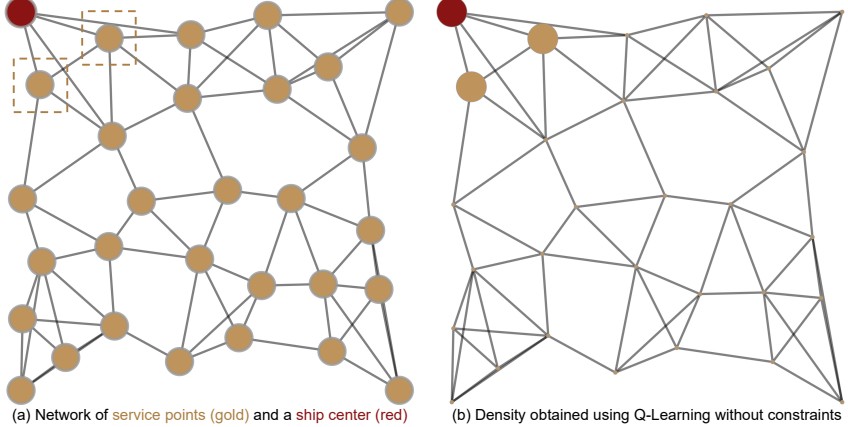

(a) Network of service points (gold) and a ship center (red)    (b) Density obtained using Q-Learning without constraints

Figure 14: An example of the express delivery service company's transportation network with one ship center and 29 service points. Left: The vans start from the service points (initial states) bounded by squares with equal probability, then visit other service points following a transition probability (policy), and finally reach the ship center (goal). Right: The standard Q-Learning method finds a policy that drives the vans directly to the goal without visiting any other service points, which minimizes the cost (traveling distance). The sizes of gold nodes represent the state density.

Table 1: Results of the express delivery service task. The maximum allowed running time to solve for a feasible policy is 600s. The cost is the expectation of traveling distance from initial states to the goal.

| Density space | Method | $\rho_{min} = 0.1$ | | | $\rho_{min} = 0.3$ | | | $\rho_{min} = 0.5$ | | |
|---|---|---|---|---|---|---|---|---|---|---|
| | | Solved | Time (s) | Cost | Solved | Time (s) | Cost | Solved | Time (s) | Cost |
| $\rho_s \in \mathbb{R}^{10}$ | CERS | True | 163.69 | 3.85 | True | 183.81 | 5.36 | True | 466.32 | 5.12 |
| | DCRL | True | 1.35 | 2.91 | True | 2.05 | 2.66 | True | 4.31 | 4.28 |
| $\rho_s \in \mathbb{R}^{20}$ | CERS | True | 227.15 | 5.80 | True | 527.26 | 6.00 | False | Timeout | - |
| | DCRL | True | 3.41 | 5.58 | True | 3.99 | 6.16 | True | 5.28 | 6.27 |
| $\rho_s \in \mathbb{R}^{100}$ | CERS | True | 161.62 | 10.26 | False | Timeout | - | False | Timeout | - |
| | DCRL | True | 3.53 | 10.23 | True | 116.24 | 12.13 | True | 153.86 | 14.06 |

formulated as the traveling distance. The frequency that each service point is visited by vans should exceed a given threshold in order to transport the packages in the service points to the ship center. Such frequency constraints can be naturally viewed as density constraints. A policy is represented as the transition probability of the vans from one point to surrounding points. The optimal policy should satisfy the density constraints and minimize the transportation distance.

Instead of comparing to methods that exert constraints on cost or value functions in the three case studies in Section 4, this case study is proposed to further understand Algorithm 1 and its key steps. In Algorithm 1 Line 5, our approach adds Lagrange multipliers to the original reward in order to compute a policy that satisfies density constraints. The update of Lagrange multipliers follows the dual ascent in Algorithm 1 Line 9 and 10, which is key to satisfying the KKT conditions. In this experiment, we try to update the Lagrange multipliers using an alternative approach and see how the performance changes. We replace the dual ascent with the cross-entropy method, where a set of Lagrange multipliers $\Sigma = [\sigma_1, \sigma_2, \sigma_3, \cdots]$ are drawn from an initial distribution $\sigma \sim Z(\sigma)$ and utilized to adjust the reward respectively, after which a set of policies $[\pi_1, \pi_2, \pi_3, \cdots]$ are obtained following the same procedure in Algorithm 1 Line 5 and 6. A small subset of $\Sigma$ whose $\pi$ has the least violation of the density constraints are chosen to compute a new distribution $Z(\sigma)$, which is utilized to sample a new $\Sigma$. The loop continues until we find a $\sigma$ whose $\pi$ completely satisfies the density constraints. We call this cross-entropy reward shaping (CERS). We experiment with 10D, 20D and 100D state spaces (corresponding to 10, 20 and 100 service points in the road network), whose density constraints lie in $\mathbb{R}^{10}$, $\mathbb{R}^{20}$ and $\mathbb{R}^{100}$ respectively. The density constraint vector $\rho_{min} : S \mapsto \mathbb{R}$ is set to identical values for each state (service point). For example, $\rho_{min} = 0.1$ indicates the minimum allowed density at each state is $0.1$. In Algorithm 1 Line 6, we use Q-Learning to update the policy for both DCRL and CERS since the state and action space are discrete.

From Table 1, there are two important observations. First, our computational time of finding the policy is significantly less than that of CERS. When $\rho_s \in \mathbb{R}^{10}$ and $\rho_{min} = 0.1$, our approach is at least 100 times faster than CERS on the same machine. When $\rho_s \in \mathbb{R}^{100}$ and $\rho_{min} = 0.5$, CERS cannot solve the problem (no policy found can completely satisfy the constraints) in the maximum allowed time (600s), while our approach can solve the problem in 153.86s. Second, the cost reached by our method is generally lower than that of CERS, which means our method can find better solutions in most cases.

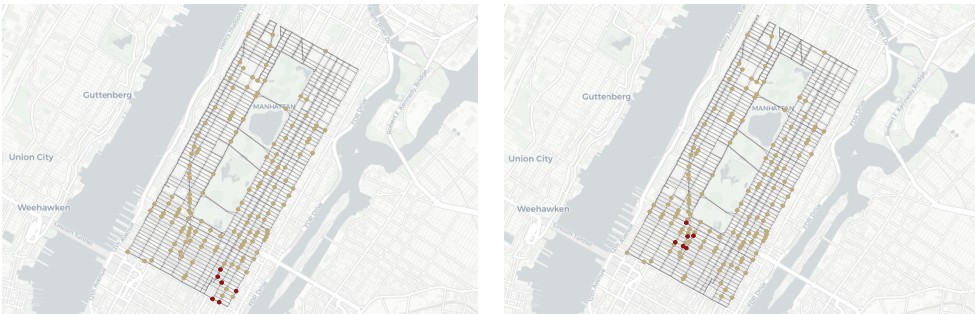

Figure 15: Autonomous electric vehicle routing in Manhattan, New York City. The objective is to control the electric vehicles to drive on the grey lines as roads and reach red nodes as goals. The left and right figures represent two scenarios where the goal sets are different. Vehicles can be charged at the gold nodes as fast charging stations. All vehicles should maintain energy levels while avoiding congesting the charge stations.

## B.4 Autonomous electric vehicle routing

In the autonomous electric vehicle routing task, we evaluate the methods in two new settings in addition to the one presented in our main paper. As is shown in Figure 15, the goal sets (red nodes) are different from Figure 4.2. The number of nodes and fast charging stations remain the same.

In Figure 16, three approaches exhibit similar performance in maintaining energy levels and avoiding running into low-energy states. Nevertheless, only DCRL succeeds in preventing the congestion in the fast charging stations by suppressing the vehicle density at each fast charging stations below the density constraints. RCPO is better than the unconstrained DDPG in terms of reducing the vehicle

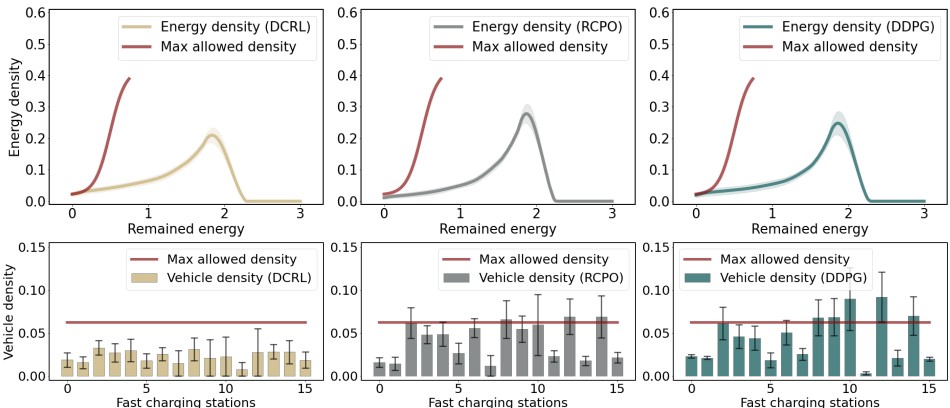

Figure 16: Density curves of the autonomous EV routing in Manhattan, with target locations shown on the left of Figure 15. The first row are the energy density and the second row are the vehicle density at each charging station using different algorithms (left to right: DCRL, RCPO, DDPG). Since there are too many charging stations, those with density lower than half of the threshold for all the three methods are omitted. Only 15 charging stations are kept in the second row. The error whiskers represent two times of the standard deviations.

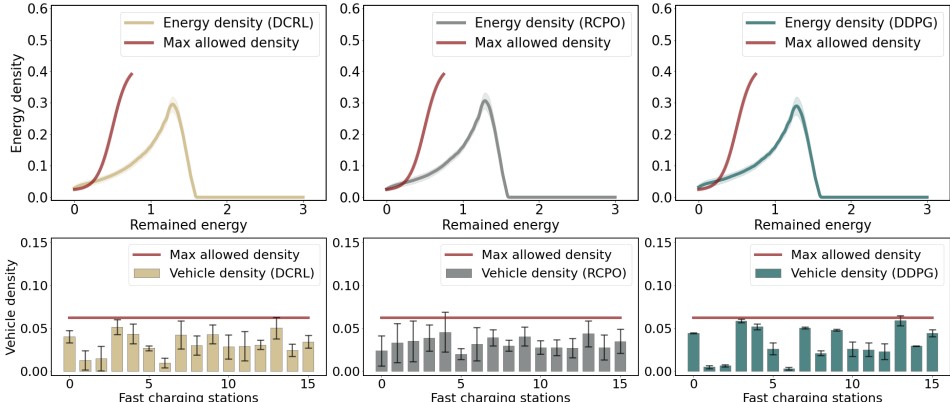

Figure 17: Density curves of the autonomous EV routing in Manhattan, with target locations shown on the right of Figure 15. The first row are the energy density and the second row are the vehicle density at each charging station using different algorithms (left to right: DCRL, RCPO, DDPG). The error whiskers represent two times of the standard deviations. Note that DDPG violates the density constraints on low-energy states.

density, but still cannot keep the density completely below the thresholds. In Figure 17, DCRL and RCPO demonstrates similar performance in avoiding low-energy states, but DDPG violates the density constraints on low-energy states. This is mostly because DDPG chooses a relatively short charging time in this scenario, which is consistent with the fact that DDPG only slightly violates the constraints on vehicle density at fast charging stations. As for the time consumption to finish the task, DCRL, RCPO and DDPG take 64.32, 60.89 and 58.40 steps on average to reach the goals shown in the left of Figure 15 and 56.31, 59.47 and 54.19 steps on average to reach the goals shown in the right of Figure 15. DDPG minimizes the traveling distance and thus requires the least amount of traveling time, while DCRL shows the best performance in satisfying the vehicle density constraints at fast charging stations.

## B.5 SAFE GYM

In this section, we experiment with the Safe Gym (Ray et al., 2019) and compare with CPO (Achiam et al., 2017) and PPO (Schulman et al., 2017) with Lagrangian (PPO-L). Both the CPO and PPO-L are implemented by Ray et al. (2019). We consider three environments in Safe Gym, including PointGoal, PointButton and PointPush. In PointGoal, the point agent aims to navigate to a goal while avoiding hazards. In PointButton, the point agent aims to press the goal button and avoid hazards. The agent will be penalized if it presses the wrong button. In PointPush, the agent is expected to

push a box to the goal position and avoid hazards. A detailed description of the environments can be found in Section 4 of Ray et al. (2019).

Figure 18: Results on Safe Gym (Ray et al., 2019) comparing to CPO (Achiam et al., 2017) and PPO (Schulman et al., 2017) with Lagrangian (PPO-L). The solid lines are the mean values of 10 independent runs. The first row shows the average return and the second row shows the constraint values. The constraint values are expected to be below the dashed lines that represent the thresholds.

Results are presented in Figure 18. In the PointGoal environment, all the three methods can satisfy the constraints, while DCRL can achieve a higher average return comparing to other methods. In the PointButton environment, CPO fails to satisfy the constraint. In the PointPush environment, it is hard for all the methods to satisfy the constraints because the pushing task is more difficult than the goal reaching task and button task. Overall, DCRL shows the highest reward and the least violation of the constraints among the three methods.

## C   DISCUSSION ON KERNEL DENSITY ESTIMATION

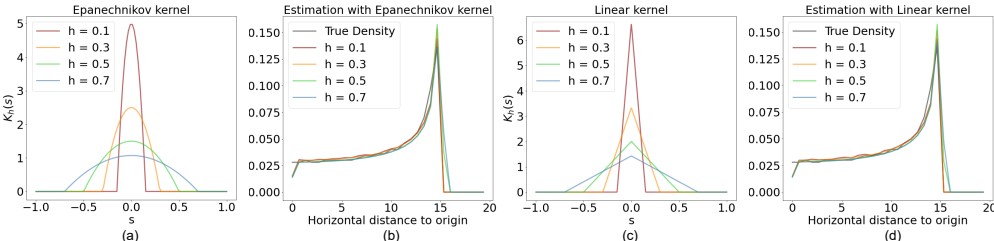

Figure 19: Empirical analysis on kernel density estimation with different kernels and bandwidths. We used the Epanechnikov kernel (visualized in (a)) and the linear kernel (visualized in (c)). We estimate the state density w.r.t. the horizontal distance to origin in the Point-Circle environment that is used in Section 4. The densities are estimated with 1000 samples.

In Section 3.3, we presented an effective way to obtain the density function using kernel density estimation when the states are continuous. In this section, we present both theoretical analysis of kernel density estimation and experimental result with different kernels and bandwidth and evaluate their influence on density estimation. There exist many results about the accuracy of kernel density estimation, we quote the following lemma from (Wasserman, 2019):

**Lemma 1.** *Let $\mathbf{h} = [h, ..., h]$ be the bandwidth of the kernel with dimension d. Let $\hat{\rho}_s$ be the kernel density estimation of $\rho_s$, then for a $\rho_s \in \Sigma(\beta, L)$, for any fixed $\delta > 0$,*

$$\mathbb{P}\left\{\sup_{x \in \mathbb{R}^n} |\hat{\rho}_s(x) - \rho_s(x)| > \Phi_+\sqrt{\frac{C \log n}{nh^d}} + \Phi_+ ch^\beta\right\} \leq \delta$$

*for some constants c and C where C depends on $\delta$.*

$$\Sigma(\beta, L) = \left\{\begin{array}{c} g : |D^s g(x) - D^s g(y)| \leq L \|x - y\|, \\ \forall |s| = \beta - 1, \forall x, y \end{array}\right\},$$

*where $D^s g(x) = \frac{\partial^{s_1 + ... + s_d}}{\partial x_1^{s_1} ... \partial x_d^{s_d}}$ and $|s| = \sum_{i=1}^d s_i$.*

This is adapted from Theorem 9 in (Wasserman, 2019), see more detail and other bias analysis results therein. Lemma 1 essentially says that if $\rho_s$ is smooth enough and one takes enough samples, the bias of the kernel density estimator can be bounded uniformly in the state space.

We used the Epanechnikov kernel and linear kernel with various bandwidths to estimate the state density w.r.t. the horizontal distance to origin in the Point-Circle environment that is used in Section 4. Results are shown in Figure 19. Although the kernels and bandwidths are different (see Figure 19 (a) and (c)), the estimated densities (see Figure 19 (b) and (d)) are all close to the true densities (grey curves).

Once the upper bound $\epsilon_\rho$ of the density estimation error is computed from Lemma 1, we can incorporate $\epsilon_\rho$ into Algorithm 1 and derive Algorithm 2, which is robust under the possibly inaccurate density estimation. In Line 9-10, we bloat the density thresholds $\rho_{max}$ and $\rho_{min}$ to $\rho_{max} - \epsilon_\rho$ and $\rho_{min} + \epsilon_\rho$, which guarantee that the density constraints will still be satisfied even when there are inaccuracies in density function estimation.

---

**Algorithm 2** Optimization with density constraints under inaccurate density estimation

---

1: **Input** MDP $\mathcal{M}$, initial condition distribution $\phi$, constraints on the state density $\rho_{max}$ and $\rho_{min}$, the upper bound $\epsilon_\rho$ of the density estimation error
2: Initialize $\pi$ randomly, $\sigma_+ \leftarrow \mathbf{0}$, $\sigma_- \leftarrow \mathbf{0}$
3: Generate experience $D_\pi \subset \{(s, a, r, s') \mid s^0 \sim \phi, a \sim \pi(s) \text{ then } r \text{ and } s' \text{ are observed}\}$
4: **Repeat**
5:     $(s, a, r, s') \leftarrow (s, a, r + \sigma_-(s) - \sigma_+(s), s')$ for each $(s, a, r, s')$ in $D_\pi$
6:     Update policy $\pi$ using $D_\pi$
7:     Generate experience $D_\pi \subset \{(s, a, r, s') \mid s^0 \sim \phi, a \sim \pi(s) \text{ then } r \text{ and } s' \text{ are observed}\}$
8:     Compute stationary density $\rho_s^\pi$ using $D_\pi$
9:     $\sigma_+ \leftarrow \max(\mathbf{0}, \sigma_+ + \alpha(\rho_s^\pi - \rho_{max} + \epsilon_\rho))$
10:     $\sigma_- \leftarrow \max(\mathbf{0}, \sigma_- + \alpha(\rho_{min} - \rho_s^\pi + \epsilon_\rho))$
11: **Until** $\sigma_+ \cdot (\rho_s^\pi - \rho_{max} + \epsilon_\rho) = 0$, $\sigma_- \cdot (\rho_{min} - \rho_s^\pi + \epsilon_\rho) = 0$ **and** $\rho_{min} \leq \rho_s^\pi \leq \rho_{max}$
12: **Return** $\pi$, $\rho_s^\pi$

---

