# OpenReview forum: "Density Constrained Reinforcement Learning"
_ICLR.cc/2021/Conference — Reject_

### Official Review · AnonReviewer4 · 2020-10-27
**Official Blind Review #3**

**Rating:** 7
**Confidence:** 2

**Review:**

The submission introduces the dual formulation of a Constrained MDP optimization. While their algorithm does not guarantee that the constraints are satisfied during learning, the experiments show that they are reasonably respected, and yield high returns.

Despite the typos and approximations I could identify, I found the exposition of the submission very clear, the arguments are convincing, and the experiments are compelling. For all these reasons, I recommend to accept it. I rate my confidence low because I have some doubts about the novelty of Theorems 1 and 2.

My main regret is the lack of discussion of the results, both on the theoretical and empirical side. Indeed:
- Theorems 1 and 2: isn't it immediate since J_d=J_p? Is it really novel?
- Proposition 1: I'm surprised the failure modes of Proposition 1 are not more discussed. How can it not converge? Can't we guarantee monotonic improvement? Likewise, how can it converge to an unfeasible solution? Can't we enforce the feasibility during the iteration? Does it only happen when no feasible solution is ever encountered?
- Experiments: what explains the apparent superiority of the primal-dual formulation as opposed to the other algorithms? What are their failure modes? Too conservative? Not safe enough? Does it depend on the setting of some hyperparameters? Isn't it simply because the constraint is defined on the density level and that Algorithm 1 is the only one designed to solve this?

Typos and minor comments:
- P_a and R_a are SxS->_
- I feel that straightening the d, like with \dd from physics package would make the integrals more readable.
- why is the density function defined as [0,\infty)xS->R_>0, isn't [0,\infty)=R_>0? What is it used for? Shouldn't it be simply S->R_>0?
- I would advise to rename \rho_s to \rho, because s is used as a variable everywhere, for instance in Eq. 2 \rho_s(s') looks like it depends on s.
- equation 3 \tho_s^\pi should be just \rho.

---

> ### Author Response · Authors · 2020-11-16
> **Response to Reviewer 4**
>
> Thanks for your positive feedback. In the following, we will address your major concerns. We have updated our draft according to suggestions by other reviewers with updates in blue color. We hope that you find our answers informative and we look forward to more discussions.
>
> **Q1: Are Theorems 1 and 2 novel?**
>
> A1: Although the duality between value function and density function is already known to the community, Theorem 1 is the first to rigorously prove that the duality also exists between Q function and density function, so it can be used on model-free RL. Theorem 2 proves that the duality still exists under constraints on state density, which is the key enabler of the proposed density constrained RL. Our work is the first to formulate density constrained RL as a more general safety constrained RL problem, on both discrete and continuous space, and use duality to solve the density constrained RL problem. Please also see our response to Reviewer 3.
>
> **Q2: How can Algorithm 1 not converge and the assumption of Proposition 1 be not satisfied?**
>
> A2: In our experiments (Section 4 and Appendix B), the algorithm could always converge to feasible solutions. The only failure mode we observed is that when the maximum and minimum density thresholds ($\rho_{max}$ and $\rho_{min}$) are extreme values that are impossible to satisfy. In this case, the perturbation terms would be updated in every iteration, yet the constraints remain unsatisfied. When setting the density thresholds, we can check whether the problem itself is feasible under such constraints and avoid infeasible settings.
>
> **Q3: What explains the superior performance of DCRL over other algorithms?**
>
> A3: Proposition 1 has proved the optimality of the proposed DCRL algorithm, which means DCRL can achieve the highest cumulative return under the constraints. Other methods would be more conservative in exploration based on the empirical observation that they did not achieve higher return than DCRL in the experiment. The constraints in Section 4.1 are all defined on value functions rather than state density. We converted the value constraints to density constraints using the duality between value function and density function. Thus, under the same constraints on value functions, the comparison is fair.

---

> > ### Comment · AnonReviewer4 · 2020-11-16
> > **Response to response Rev 4**
> >
> > Thank you for your clarifications. They are helpful. I am still a bit unsure about the novelty, and since Reviewer 3 raised the same concern in a more affirmative way, I am going to pay attention to his response to your argumentation.

---

### Official Review · AnonReviewer1 · 2020-10-29
**Novel perspective on constrained RL, with modest improvements**

**Rating:** 7
**Confidence:** 3

**Review:**

Summary:
This paper proposes to solve constrained RL through the lens of setting constraints on state density, rather than setting constraints on value functions. The main contributions are (1) a proof of the duality between the density function and Q-value function in constrained RL, and (2) a primal-dual algorithm, called DCRL, for solving the constrained density problem.

Recommendation:
This paper tackles an important problem, and presents a theoretically-grounded novel perspective on constrained RL. I'm leaning toward accept, but have several suggestions regarding the empirical evaluation, that are detailed below.

Pros:
* The paper is clearly written and well-motivated.
* It seems easier to express constraints in terms of the state density function rather than Q-values.
* The empirical evaluation contains a comparison to several state-of-the-art approaches for constrained RL.
* DCRL shows modest improvements in the experiments, compared to the baselines.

Cons:
* I would like to see a comparison against the simple yet effective baseline of PPO combined with Lagrangian relaxation, with respect to constraints on Q-values.
* I would like to see results on more challenging constrained RL tasks, where the state density function is harder to estimate, for instance the Safety Gym task suite [1]. These tasks are particularly interesting because CPO (surprisingly) performs worse on them than the simpler PPO-Lagrangian approach. I'm curious whether DCRL would struggle as well.

Comments/questions regarding notation:
* Using $P_a(s, s')$ and $R_a(s, s')$ in the definition of an MDP (Section 2) is odd; I recommend using the typical notation of $P(s, a, s')$ and $R(s, a, s')$.
* In the definition of MDPs, $V^\pi(s)$ is incorrectly referred to as _reward_ / _total reward_, but it should instead be referred to as the _return_ or _expected cumulative discounted reward_.
* It's not clear that the $[0, \infty]$ for the density function $\rho$ corresponds to $t$ (in the first sentence of the Density Functions section).
* Why are there two $s$'s in $\rho_s^\pi(s)$ (e.g., in Equations 1 and 2). Should the subscript be a $\gamma$ instead?
* What does it mean to "linearly interpolate" $\sigma_+$ and $\sigma_-$ (at the end of Section 3.3)?

[1] Ray et al. Benchmarking Safe Exploration in Deep Reinforcement Learning. 2019.

---

> ### Author Response · Authors · 2020-11-16
> **Response to Reviewer 1**
>
> Thanks for your detailed comments. In the following, we will address your major concerns. We have added your requested comparison experiments and updated our draft accordingly with updates in blue color. We hope that you find the updated information erasing your concerns.
>
> **Q1: Add comparison to PPO with Lagrangian relaxation. Add experiment on Safe Gym.**
>
> A1: We have added the experiments in **Appendix B.5** on the Safe Gym and compared to PPO with Lagrangian relaxation (PPO-L). Three tasks are considered, including PointGoal, PointButton and PointPush. It is shown that DCRL has better performance than PPO-L on improving the reward while respecting the constraints. In fact, RL with constrained value functions is a special case of our proposed density constrained RL since we can define the density constraint in the form of a value function.
>
> **Q2: Some notations are old. $V^{\pi}(s)$ is sometimes incorrectly referred to as reward or total reward.**
>
> A2: We have changed the notations and refer to $V^{\pi}(s)$ as expected cumulative discounted reward. The changes can be found in the updated draft.
>
> **Q3: It is not clear that $[0, \infty]$ corresponds to the time $t$ in the definition of $\rho$.**
>
> A3: It is true that $[0, \infty]$ corresponds to $t$. But in our paper, we mainly consider the stationary density $\rho_s(s)$ that does not depend on $t$. To avoid confusion, we have removed the $[0, \infty]$ in our definition of density.
>
> **Q4: Why are there 2 $s$’s in density $\rho_s^{\pi}(s)$?**
>
> A4: The subscript s means the density is stationary.
>
> **Q5: What does it mean to linearly interpolate $\bar{\sigma}_{+}$** **and** $\bar{\sigma}_{-}$?
>
> A5: $\bar{\sigma}$  only contains the values of Lagrangian multipliers on finite state samples.  When evaluating $\bar{\sigma}$ on continuous states, e.g., state $s$, we perform linear interpolation on the Lagrangian multipliers on samples near s so that the value of $\bar{\sigma}$ at $s$ can be evaluated.

---

> > ### Comment · AnonReviewer1 · 2020-11-18
> > **Response to Authors**
> >
> > Thank you for the answers to my questions, and for the extra experiments! (Minor typo in Appendix B.5—the name of the task suite is "Safety Gym", rather than "Safe Gym".)
> >
> > The inclusion of the Safety Gym experiments (and the PPO-Lagrangian baseline) significantly strengths the empirical evaluation in my view, and I've increased my score to an accept. I still think the improvement in task performance that DCRL obtains is relatively small, but at least this small improvement is consistent across many different tasks.

---

### Official Review · AnonReviewer3 · 2020-10-29
**Concerning the novelty**

**Rating:** 5
**Confidence:** 4

**Review:**

This paper studied density constrained reinforcement learning (DCRL), which compared with standard RL has constraints on the stationary state distribution. It proposes a primal-dual optimization method and proves its optimality.


The theoretical analysis in this paper assumes $
P_a(s^\prime, s)$ is perfectly known whenever needed.  In fact, in this case formulations (3), (4), (5), (6) are simple deterministic convex (linear) optimization (regardless if density constraints are introduced or not).  The duality $J_d$ and $J_p$ duality follows directly from duality for convex programs.   The developed algorithm, Algorithm 1, is also simple application of (primal)-dual subgradient method for convex programs. If the D_s update and  \rho_s^\pi update are done exactly, i.e., assuming perfect knowledge of  P_a(s^\prime, s), then the theorems in this paper is just simple consequence of well-established convex optimization results. (I looked into the proof in the appendix. It seems the analysis indeed assumes perfect $P_a(s^\prime, s)$ in the analysis.)

Overall, I think the technical contribution and novelty of this are marginal.

---

> ### Author Response · Authors · 2020-11-16
> **Response to Reviewer 3**
>
> Thank you for raising your concerns on the novelty of our paper. We would like to point out a couple of misunderstandings which may lead to your concern of novelty. We have also updated our draft to make both the theoretical guarantee and experiments results stronger as suggested by other reviewers.
>
>
> **First**, we would like to clarify that we don’t assume the transition probabilities to be known. Instead, both the primal update and dual update are totally model-free. The primal update uses standard RL algorithms by taking samples from the environment; the dual update uses the samples from the primal step to approximate the density function (see our answers to Reviewer 2 on how to approximate the density function) and update the perturbation term. All calculation is done model-free, relying only on the samples. We do not assume the perfect knowledge of $P_a(s’, s)$.
>
>
> **Second**, although duality is a standard approach in convex programs, we argue that our work is non-trivial and important, as also agreed by other reviewers. The fact that duality is a standard technique in convex programs does not mean that it is obviously known how to use it to solve density/safety constrained RL. The duality relationship has inspired many important works over decades. We feel it is unfair to use the fact that such duality is a standard technique to conclude that our work is incremental.
>
>
> **To be more concrete about our contributions:** 1) Our work is the first to formulate density constrained RL as a more general safety constrained RL problem, on both discrete and continuous space. We are also the first to rigorously prove and use the duality between density functions and Q functions over continuous state space to solve density constrained RL. Our algorithm can automatically solve density constrained RL with guarantees on satisfying the density constraints, unlike the “reward-based” RL which needs fine-tuning and trial-and-error for each use case. 2) Any nonnegative dual variable satisfying the conservation law (Liouville PDE in the continuous case) is a valid dual. However, among all valid dual variables, the state density we defined is associated with a clear physical interpretation as the concentration of states, and we are able to directly apply constraints on the density in RL. 3) we accomplish the non-trivial work of using the algorithm on an extensive set of case studies and show its advantage over other state-of-the-art safety constrained RL methods.
>
>
> Regarding concerns on the robustness of the approach using $P_a$ from samples but not perfectly known, there is plenty of literature on this topic yet the robustness relies on the implementation. For example, a discrete MDP is much easier to analyze than a continuous MDP where function approximation is used. We plan to give a rigorous robustness analysis in our future work.

---

> > ### Comment · AnonReviewer3 · 2020-11-19
> > **Response to authors**
> >
> > Let me clarify what I mean by "The theoretical analysis in this paper assumes $P_a(s, s^\prime)$ is perfectly known whenever needed"
> > (1) The duality part follows from the duality for "constrained convex program", which is theoretically available under $P_a(s, s^\prime)$.  Do you agree with this?
> > (2) The thing that matters for the "model-free" setting is the practical convergence of Algorithm 1 (and the newly added Algorithm 2). Unfortunately, there is no "convergence analysis" provided.   Proposition 1 is rather meaningless. It states that "if Algorithm 1 converges to a feasible solution *that satisfies the KKT condition*, then it is optimal...".  How can you ensure the "if" part?  The KKT condition is only well defined with the transition prob.  Suppose you have a KKT point, the optimality again trivially follows from the standard convex optimization theory.
> >
> > I saw that the authors presented a new algorithm (Alg2) that uses inaccurate density estimation. But its convergence is still not provided.  I guess the authors may think the convergence of Alg1/Alg2 follows directly from its "deterministic convex opt" form, i.e., dual-subgradient method.   This is not true for the claimed "model-free" setting. (The convergence is only obvious when the  transition is perfectly known or under Rev2's interpretation that "All the derivation and the stop criterion is based on infinite number of samples".  In fact, to be more rigorous, dual subgradient has convergence with constant step size \alpha only when the obj is strongly convex. Otherwise, it converges to an \epsilon(\alpha) optimal point.)  The new lemma (Lemma 1) might be useful to analyze the convergence because it can be potentially transformed to error bounds on the "dual subgradient".   That being said, I still consider the technical contribution of the current version relatively low.

---

> > > ### Author Response · Authors · 2020-11-20
> > > **Response to reviewer - Part 2**
> > >
> > > ##### Following [Part 1](https://openreview.net/forum?id=jMc7DlflrMC&noteId=uXb6kaLlsfl)
> > >
> > > In terms of empirical evaluation of convergence, through our extensive experiments in Section 4 and Appendix B, including MuJoCo tasks, autonomous vehicle routing, electrical motor control, drone applications, and Safety Gym, we always observe that Algorithm 1 could converge when the problem is feasible. The empirical results demonstrate that the DCRL method not only successfully enforces the constraints, but also achieves higher cumulative reward than baseline methods.
> > >
> > > We also want to clarify that our major contribution in this paper is indeed not the theoretical proof of the duality between density and Q-function, but rather the first to formulate the density constrained RL problem, which has clear practical safety meanings, and use an extensive and diverse set of benchmarks to show our method’s empirical advantages. We will modify the paper to make sure this part is clearer. Proving convergence of primal-dual infinite-dimensional constrained optimization with RL as the primal update is indeed a hard problem and we aim to solve it in the near future. However, we hope that we are not punished too much because we believe that we proposed a novel formulation of safe RL problems and have made solid contributions in addressing this problem both theoretically and empirically in this paper. We sincerely hope that you can re-evaluate our paper based on these contributions.
> > >
> > > Our contributions have also been appreciated by other viewers. For example, Reviewer 2 points out that we demonstrated promising empirical results compared to other baselines, and believes applying the duality between Q-functions and density functions to solve constraint RL is new and helpful in safe RL. Reviewer 1 considers the density constrained RL as a novel perspective, and also agrees that expressing constraints using the state density is easier than using the Q-values. Reviewer 1 also points out that the updated Safety Gym experiments significantly strengthen the evaluation and our performance is consistent across many different tasks.
> > >
> > > [1] Nedić, Angelia, and Asuman Ozdaglar. "Subgradient methods for saddle-point problems." Journal of optimization theory and applications 142.1 (2009): 205-228.
> > >
> > > [2] He, Bingsheng, and Xiaoming Yuan. "Convergence analysis of primal-dual algorithms for a saddle-point problem: from contraction perspective." SIAM Journal on Imaging Sciences 5.1 (2012): 119-149.
> > >
> > > [3] Ding, Dongsheng, et al. "Natural Policy Gradient Primal-Dual Method for Constrained Markov Decision Processes." Advances in Neural Information Processing Systems 33 (2020).

---

> > > ### Author Response · Authors · 2020-11-20
> > > **Response to reviewer - Part 1**
> > >
> > > We would like to thank the reviewer for the valuable comments on convergence analysis. We hope the reviewer finds the following response helpful for addressing the concerns.
> > >
> > > Q1: The duality part follows from the duality for "constrained convex program", which is theoretically available under $P_a(s,s’)$. Do you agree with this?
> > >
> > > A1: We partially agree with your comment but we do not think that using $P_a(s,s’)$ in the theoretical analysis is causing any practical issue. To use an analogy, the standard RL method is built on dynamic programming and the Bellman operator, which, similar to your argument, depends on the transition probabilities $P_a(s,s’)$. However, people normally do not say that the standard RL algorithms rely on the transition probabilities being perfectly known. Our case is the same as in standard RL. We use $P_a(s,s’)$ in our analysis to show the duality and in the algorithm derivation, yet the algorithm does not use the transition probabilities anywhere and is completely model-free. Honestly, we do not know a way to claim we do not have any information about $P_a(s,s’)$ and still perform the theoretical analysis. Please advise if you have a better idea.
> > >
> > > Q2:Proposition 1 is rather meaningless
> > >
> > > A2: We want to clarify that by KKT condition we meant the KKT condition for the outer-loop. The inner-loop is a perturbed unconstrained reinforcement learning problem. Proposition 1 showed that when the density function (primal variable of the outer-loop) and the Q function (dual variable together with the perturbation) satisfy the KKT condition, the solution of the perturbed unconstrained RL is the optimal solution to the constrained RL, which is needed to establish the optimality of our algorithm.
> > >
> > > Q3: Question on the convergence analysis of Algorithm 1.
> > >
> > > A3: We acknowledge that convergence analysis is desired for the primal-dual algorithm, which we never thought was trivial. In fact, we did spend considerable time on the convergence proof. However, note that the proposed primal-dual algorithm solves an infinite-dimensional constrained linear programming, and the primal update of Algorithm 1 solves the model-free RL (Line 6) using function approximation. The well-known convergence proof of primal-dual algorithm typically depends on gradient or subgradient-based methods [1] or proximal point methods [2], yet the primal RL update step is technically not a subgradient update or a proximal point update. In the case of finite-state MDP, one can prove the convergence of the primal-dual algorithm[3], but it is probably easier to call an LP solver to solve such a problem as density-constrained optimization problems instead of using primal-dual approaches. In the case of primal-dual infinite-dimensional constrained optimization with RL as the primal update, the convergence proof could be extremely challenging, and we have claimed in the Discussion section that this will be our future work.
> > >
> > > **See part 2 for the remaining response and the references**

---

### Official Review · AnonReviewer2 · 2020-11-04
**Review for Density Constrained RL**

**Rating:** 6
**Confidence:** 4

**Review:**

In this paper, the authors explored the duality between density function and value function in the setting of density constrained RL. Based on this, the author proposed a new safe constrained policy optimization algorithm by jointing optimizing policy and lagrangian multipliers of the density constraints, with an extra effort to estimate the stationary state density of current policy $\pi$ using kernel density estimation. The empirical results demonstrate that the proposed algorithm  is effective in several mujoco benchmark and  autonomous electric vehicle controlling.  I believe exploring the duality between density function and value function in the  setting in density constrained reinforcement learning is new, and may be helpful in safe reinforcement learning.

Followings are my detailed questions and comments:

- I have a major concern about estimating the marginal (or stationary) state function of the policy $\pi$. The author use kernel density estimation to estimate $\rho(s)$, which I think deserve more discussion in this paper, and at least some empirical experiments should be conducted to verify its accuracy. Moreover, since the estimation requires to use kernels, it would be good to see the empirical results of choices of kernels and bandwidth of the kernel.

- All the derivation and the stop criterion is based on infinite number of samples, which will not guarantee that the constrain is satisfied using finite mc samples to estimate (since you don't have the transition, you can only use samples to estimate). Will there be any guarantee that the exact estimation of density and constrain is upper bounded by some empirical estimation, such that we can use mc sampling to estimate the density or constraints?

- The duality between the density function and value function is well-known in the community, and has drawn great attention recently in the policy evaluation community [1, 2, 3, 4], which I think the authors should have some discussion on this in the related work,  and to see if the techniques can be used to estimate the density functions.

Overall I think the paper proposed a new perspective for density constrained rl, which I think is interesting and is publishable if my above concerns are well addressed.

--------------------
I will update my scores if my questions are addressed:)

[1] Nachum, Ofir, et al. "Dualdice: Behavior-agnostic estimation of discounted stationary distribution corrections." Advances in Neural Information Processing Systems. 2019.

[2] Tang, Ziyang, et al. "Doubly robust bias reduction in infinite horizon off-policy estimation.", ICLR 2020.

[3] Uehara, Masatoshi, Jiawei Huang, and Nan Jiang. "Minimax weight and q-function learning for off-policy evaluation." arXiv preprint arXiv:1910.12809 (2019).

[4] Nachum, Ofir, and Bo Dai. "Reinforcement learning via fenchel-rockafellar duality." arXiv preprint arXiv:2001.01866 (2020).

---

> ### Author Response · Authors · 2020-11-16
> **Response to Reviewer 2**
>
> Thanks for your detailed comments. In the following, we will address your major concerns. We have updated our draft accordingly with updates in blue color. We hope that you find the updated information erasing your concerns.
>
> **Q1: More discussion on the kernel density estimation.**
>
> A1: We added **Appendix C**, in which we have added both the theoretical upper bounds and the empirical results of the estimation error of using kernel density estimation. The theoretical bounds are from well-known results. In the experiments, we chose the  Epanechnikov kernel and linear kernel with 4 different bandwidth (8 configurations in total), and evaluated their accuracy on estimating the state density of the Point-Circle task. It is shown that all the 8 configurations can result in a good estimation of the state density.
>
> **Q2: Guarantees of the algorithm and theory using a finite number of samples to estimate the density.**
>
> A2: We have also addressed this in our newly added Appendix C. In short, the guarantee can be easily computed using the theoretical kernel density estimation accuracy found and then added as a buffer to the density constraint to be robust against the errors of density estimation. We have provided Algorithm 2 in Appendix C as an updated version of Algorithm 1, to demonstrate how to consider the error in density estimation when using our proposed DCRL approach.
>
> **Q3: Related works on the duality between the density function and value function.**
>
> A3: Thank you for pointing out the references. We agree that these are very relevant papers and we have added a discussion on the related work, see the highlighted part of Section 1.

---

> > ### Comment · AnonReviewer2 · 2020-11-23
> > **Thank you for your clarification**
> >
> > I thank the authors for empirical evaluation of using kernel density estimation to estimate the stationary state distribution. For Q2, the guarantee of kde estimation is easy.
> >
> > After carefully reading other reviewers' comments, I have a major concern about the convergence of Algorithm 1, since the update of $\sigma^{-}$ and $\sigma^{+}$ part is tricky, which is derived from the KKT condition. So it is hard to understand whether the algorithm will final converge.
> >
> > Since the empirical validation is promising, and more precise analysis or update formula may be done as  future works, I will keep my score 6.

---

> > > ### Author Response · Authors · 2020-11-23
> > > **Response to Reviewer 2**
> > >
> > > Thank you for your response. We are wondering whether you can elaborate more on your concern about the update of $\sigma_+$ and $\sigma_-$? We thought it was  natural to derive the primal-dual algorithm based on the KKT condition since our problem is convex-concave.
> > >
> > > Regarding the convergence proof, we acknowledge that it is important and we have spent a lot of time working on the proof. We would like to first point out that the density constrained RL in (3) is a convex-concave problem, meaning that standard primal-dual algorithm such as the subgradient method with diminishing step size is guaranteed to converge when the dual update can achieve an optimal solution. However, since in our implementation, the dual update (i.e., the RL with perturbation) is done with a neural network for function approximation, it is technically not a gradient/subgradient method or a proximal point method update. It is well-known that RL convergence with deep NN is hard to prove, and that is exactly the obstacle we are facing. We could assume that the optimal solution is obtained at  the dual update and proceed to give a convergence proof as in [1], but we feel it is not precisely reflecting the  practice. We feel that such a proof cannot give us insights on the empirical results. Therefore, similar to many deep RL works, we focused our energy on showing the empirical convergence and that our method outperforms state-of-the-art safe RL methods. We would like to work out the nontrivial convergence proof of the real implementation using NN in RL as our important future work, which would also have a significant impact in the deep RL community as a standalone contribution.
> > >
> > > [1] Paternain, Santiago, Luiz Chamon, Miguel Calvo-Fullana, and Alejandro Ribeiro. "Constrained reinforcement learning has zero duality gap." In Advances in Neural Information Processing Systems, pp. 7555-7565. 2019.

---

### Author Response · Authors · 2020-11-25
**General remark after rebuttal**

We are sincerely grateful to the reviewers for their valuable comments and suggestions. We also appreciate that 3 out of 4 reviewers gave us very positive evaluations. One reviewer also raised the score during rebuttal after we added more experiments, which “significantly strengths the empirical evaluation”. We understand that reviewers 3 and 2 have concerns about the convergence. However, as we have pointed out in the response, proving the convergence of an algorithm with deep RL inner-loop is a very well known hard problem. On the other side, proving the convergence of our algorithm with assumptions on oracles is already discussed in many related papers and does not provide useful guidance and insights to our empirical experiments. More importantly, we have used extensive experiments to show that when the problem itself is feasible, our Algorithm 1 could always converge. We want to emphasize that our major contribution is the novel formulation of density constrained RL as a more general safety constrained RL problem and the first to use duality between density functions and Q functions over continuous state space to solve density constrained RL. Our method has demonstrated consistent improvements over many tasks in terms of satisfying the constraints and achieving high rewards.

We really appreciate the reviewers’ suggestions that helped us significantly improve the quality of our submission. We hope the area-chair can recognize our contribution and novelty which have been well-acknowledged by the positive reviews as “proposed a new perspective for constrained RL”, “clearly written”, “well-motivated”, and the “improvement is consistent across many different tasks”.

---

### Decision · Program_Chairs · 2021-01-07
**Final Decision**

**Decision:**

Reject

**Comment:**

The reviewers raised several theoretical and empirical questions about the paper. During the rebuttals, the authors seem to  successfully address the experimental issues, in particular those raised by Reviewers 1 and 2. However, the theoretical concerns have mainly remained unanswered. Reviewer 2 has a major concern about the convergence of Algorithm 1, both Reviewers 3 and 4 (in particular Reviewer 3) have several questions about the theoretical analysis and see the technical contribution of the work relatively low. These are not minor issues and require a major revision to be properly addressed. So, I suggest that the authors take the reviewers comments into account, revise their work and properly address the issues raised by the reviewers, and prepare their work for an upcoming conference.